# Bayesian joint inference of hydrological and generalized error models with the enforcement of Total Laws

Mario R. Hernández-López<sup>1</sup>, Félix Francés<sup>1</sup>

<sup>1</sup> Research Institute of Water and Environmental Engineering, Universitat Politècnica de València, Spain

Correspondence to: Mario R. (maherlo@hma.upv.es)

# Abstract.

Over the years, the Standard Least Squares (SLS) has been the most commonly adopted criterion for the calibration of hydrological models, despite the fact that they generally do not fulfill the assumptions made by the SLS method: very often errors are autocorrelated, heteroscedastic, biased and/or non-Gaussian. Similarly to recent papers, which suggest more appropriate models for the errors in hydrological modeling, this paper addresses the challenging problem of jointly estimate hydrological and error model parameters (joint inference) in a Bayesian framework, trying to solve some of the problems found in previous related researches. This paper performs a Bayesian joint inference through the application of different inference models, as the known SLS or WLS and the new GL++ and GL++Bias error models. These inferences were carried out on two lumped hydrological models which were forced with daily hydrometeorological data from a basin of the MOPEX project. The main finding of this paper is that a joint inference, to be statistically correct, must take into account the joint probability distribution of the state variable to be predicted and its deviation from the observations (the errors). Consequently, the relationship between the marginal and conditional distributions of this joint distribution must be taken into account in the inference process. This relation is defined by two general statistical expressions called the Total Laws (TLs): the Total Expectation and the Total Variance Laws. Only simple error models, as SLS, do not explicitly need the TLs implementation. An important consequence of the TLs enforcement is the reduction of the degrees of freedom in the inference problem namely, the reduction of the parameter space dimension. This research demonstrates that non-fulfillment of TLs produces incorrect error and hydrological parameter estimates and unreliable predictive distributions. The target of a (joint) inference must be fulfilling the error model hypotheses rather than to achieve the better fitting to the observations. Consequently, for a given hydrological model, the resulting performance of the prediction, the reliability of its predictive uncertainty, as well as the robustness of the parameter estimates, will be exclusively conditioned by the degree in which errors fulfill the error model hypotheses.

# 1 1. Introduction

In hydrologic simulation, many parameters are not directly observable or easily inferred from measured data. For this reason, these parameters must be estimated indirectly by an inverse process (also called calibration) that conditions the parameter estimates and the model response on historically observed input-output data (Pokhrel and Gupta, 2010). Model parameter inferences are based on a likelihood function which quantifies the probability that the observed data were generated by a particular parameter set (Box and Tiao, 1992). Over the years, the Standard Least Squares (SLS) criterion has been the most commonly adopted method for the calibration of the unknown parameters in hydrological modeling, although it has long been known that errors of hydrologic models are generally autocorrelated and heteroscedastic (Sorooshian and Dracup, 1980) not fulfilling the underlying hypotheses to SLS. Also, model inputs and model structural errors are important sources of uncertainty and they are ignored by the SLS scheme. I.e., the SLS parameter estimates are adjusted with a particular calibration data set to partially compensate the input errors (Kavetski et al., 2003; Schoups and Vrugt, 2010) and/or the structural errors (Schoups and Vrugt, 2010) yielding a set of biased parameters.

13 The present paper is framed as a formal Bayesian aggregated approach to characterize the parameter and total predictive 14 uncertainty, with a direct method for handling the bias, autocorrelation, heteroscedasticity and non-normality of the 15 hydrological model errors. This approach follows to that proposed in Schoups and Vrugt (2010), but in present paper we have 16 introduced significant improvements. Schoups and Vrugt (2010) proposed a formal likelihood function based on a general 17 error model (called GL error model) that allows for model bias and for autocorrelation, non-stationarity, and non-normality 18 of model errors. This formal approach preserved the advantages of being developed from a theoretical basis and having the 19 possibility of checking the assumptions, while at the same time it improved flexibility and reduced the need for unrealistic 20 assumptions about the errors. That study was based on the joint inference of both the hydrological and error model 21 parameters (hereinafter joint inference). The main contribution of that research was the treatment of the error 22 heteroscedasticity and non-normality with a direct method instead of using a transformational one, as it had been previously 23 used in several case studies (Bates and Campbell, 2001; Kuczera, 1983; Reichert and Mieleitner, 2009; Vrugt et al., 2009b). 24 However, according to Schoups and Vrugt (2010), something in the method seemed not to work properly: the analysis 25 performed with heteroscedasticity, autocorrelation, and non-normality error parameters for the second case study (Guadalupe 26 River basin) gave results with "large and meaningless" prediction uncertainty bands. These authors pointed out that the 27 reason was the large inferred value for the autocorrelation coefficient. To avoid this problem, the authors decided to adjust 28 the autocorrelation parameter to an observed sampling value, rather than inferring it jointly and automatically.

Evin et al. (2013) described the challenges of fitting hydrological model parameters jointly with the autocorrelation and heteroscedasticity parameters of error models, and found a solution to the aforementioned problem in Schoups and Vrugt (2010). The key to the problem solution depended on how the autoregressive model was applied. Applying it directly to the errors can produce instability in the computations and, as a result, a poor error model with large predictive uncertainty. This instability could be avoided by applying the autoregressive process to the studentized errors. Thus, the heteroscedasticity model was performed on errors (instead on innovations) and the de-correlation process was applied on the previously variance-stabilized errors.

In a useful and comprehensive research on the comparison of inference methodologies, Evin et al. (2014) detected that problems remained in the joint estimation of both hydrological and error model parameters. They concluded that the joint inference could be non-robust due to multiway interactions between the hydrological parameters (related with the water balance in their case study) and the error model parameters, particularly, heteroscedasticity and autocorrelation error model parameters.

Moreover, several related researches have avoided a full joint inference either without the modeling of the error's autocorrelation (Cheng et al., 2014; Westra et al., 2014) or using a transformational (or indirect) method for the treatment of the error heteroscedasticity (Bates and Campbell, 2001; Cheng et al., 2014; Del Giudice et al., 2013; Kuczera, 1983; Reichert and Mieleitner, 2009; Vrugt et al., 2009b). Even there are authors who combine the Markov Chain Monte Carlo (MCMC) sampling of the hydrological parameters with a maximum likelihood estimation of the error model parameters conditioned on each MCMC hydrological parameter sample (Zheng and Han, 2015; Han and Zheng, 2016), making a sort of pseudo-joint

inference. However other authors as Scharnagl et al.(2015) tried the joint inference without success, finding similar problems
 to those exposed in Schoups and Vrugt (2010) or Evin et al. (2013, 2014).

Consequently, the general purpose of our research is to find out the reasons and to overcome the drawbacks identified in the

previous studies, defending the reliability of the Bayesian joint inference methodology. This defense is necessary because

non-using the joint inference involves a range of problems (Evin et al., 2014): i) the use of a faulty error model when 52 calibrating the hydrological parameters produces biased parameter estimates and an incorrect parameter and predictive

uncertainty estimates; ii) the estimation of error model parameters conditioned on a particular estimate of hydrological

parameters (assuming that they could be biased) ignores interactions between both sets of parameters, and may produce an 55 incorrect estimation of the predictive uncertainty.

The specific objectives covered by this paper are the following. Firstly, we have improved the formal general likelihood function for parameter and predictive inference of hydrological models developed by Schoups and Vrugt (2010), implementing the previously mentioned recommendation proposed in Evin et al. (2013). Secondly, this paper seeks to shed some light on the causes of the incorrect behavior of the joint inference which has been detected in Schoups and Vrugt (2010), Evin et al. (2013, 2014) and Scharnagl et al. (2015).

Todini (2007) pointed out that "(...) *uncertainty* (...) *is 'conditional' upon the forecasting model prediction*". That idea is the basis to understand that the concept of predictive uncertainty must be linked with the conditional distribution of a predicted

variable given its related model prediction namely  $p(y|y_s)$ . Therefore, the existence of that conditional distribution

implies also the existence of its joint distribution  $p(y, y_s)$ , which is the foundation of the general framework proposed in

this paper. When we define the conditional distributions through any of its statistical features (i.e. shape, position, etc.), it is important to note that we are defining in parts their parent joint distribution. So, these parts must comply with the necessary conditions/restrictions to conform all together such joint distribution. In particular, two necessary conditions are the Total Variance Law and the Total Expectation Law (for details about these laws see for example Blitzstein and Hwang, 2014), which relate the marginal and conditional distributions of a given joint distribution. This paper enforces both Total Laws to the heteroscedasticity and non-constant bias models, since they are part of the conditional distributions modeling and, hence,

they belong to the joint probability density function (hereinafter pdf). The presented case studies will show how not enforcing

the Total Laws could be the origin of the previously found problems with the Bayesian joint inference.

The next section presents the generalized error statistical model followed by this research. "General" in the sense that error 74 model permits the possibility for the errors to be autocorrelated, heteroscedastic, biased and/or non-Gaussian. Section 3 deals 75 with the joint Bayesian parameter inference. A new formal generalized likelihood function is presented and the methods for 76 obtaining the posterior of parameters and the predictive distribution are outlined. Section 4 describes the different inference 77 settings and how to apply the Total Laws on them. Section 5 applies the exposed methodology to estimate the parameter and 78 predictive uncertainty, through the joint inference with several error models and two lumped hydrological daily models, using 79 hydrometeorological data from a basin in the Model Parameter Estimation Experiment (MOPEX) data set (Duan et al., 2006). 80 The fulfillment of assumptions in the hypothesized error models, as well as several indicators of performance will be tested. 81 Section 6 discusses the results and section 7 summarizes our findings.

# 82 2. Generalized error statistical model

# 83 2.1 Why do we need a reliable error model?

Let us consider a random variable of hydrological predictive interest to be forecasted, for which the observations are made and which will be called the predictand y (e.g. the streamflow at a catchment outlet q, or any other state variable of interest). The predictand y can be sampled at N constant time steps and it is jointly observed and sampled with a set of predictors or model inputs, **X** (e.g. precipitation, temperature, etc.). This research will consider **X** as deterministic

variables, which are able to explain "something" about the predictand's behavior.

Let us also consider any model output  $y_s$  (e.g. the simulated streamflow  $q_s$ , or some other observable and simulated state

variable) as a random variable; not because a stochastic model is used (which is not our case), but by the fact that we can 91 observe that these simulated variables generally do not match their observations. In this sense, Todini (2007) pointed out that 92 "(...) a scatter will always be observed in the q-qs plane (...) a representation of the joint sample frequency of q and qs that 93 can be used to estimate its joint probability density". That is to say, there is an inherent uncertainty whose origin is in the 94 model structure soundness, in the observed data errors of predictand and predictors, and ultimately in the inherent 95 unpredictability in deterministic terms of the natural phenomena. Montanari and Koutsoyiannis (2012) concluded that 96 uncertainty is unavoidable in Hydrology so it is impossible to produce a fully deterministic model that would eliminate the 97 uncertainty, and modeling schemes need to explicitly recognize its role.

As stated in Todini (2007), two different objectives can be differentiated in hydrological modeling: parameter estimation and 99 hydrological prediction. Parameter estimation is the procedure for obtaining parameter values with a physical meaning, which 100 help us to understand the nature of the modeled processes. That is to say, the aim of the hydrological parameter estimation is 101 not (it should not be) to get the best fit with the observations, but achieving the most plausible estimates with the best 102 possible error model for the given observed data set. Hydrologists should assume the fact that, given a correct error model 103 (actually a correct likelihood function), the most plausible parameter set can produce simulated results which show a poor fit 104 to the observations, even in the calibration period. In other words, an inference with the correct error model yields the most 105 plausible parameter set, which will not necessarily yield the best fit to the observed data, but will produce the best possible 106 result, taking into account the limitations of the hydrological model and the observational errors. In a classical Frequentist 107 context, the most plausible parameters provide the Maximum Likelihood and in a Bayesian framework are the Maximum a 108 Posteriori (from now MAP).

- With regard to the problem of making predictions about some variable of interest, the correct estimation of the hydrological
- parameters is not necessary, since the predictive uncertainty,  $p(y|y_s)$ , can be obtained through the direct modeling of the
- joint distribution of the predict and its related model output random variable, namely the  $p(y, y_s)$ , as it was first shown

by Krzysztofowic (1999) and many others later. Montanari and Brath (2004), under certain assumptions, carried out the

predictive uncertainty assessment through the modeling of the joint distribution of the model errors and the model predictions

$p(e, y_s)$  instead of modeling  $p(y, y_s)$ , although all of them formulated the problem by using a meta-Gaussian model

(Kelly and Krzysztofowicz, 1997).

When we want to meet both objectives parameter estimation and predictive uncertainty assessment at the same time, we can address the issue through the modeling of the arising uncertainty from the uncertainty sources. For this task we can use two 117 118 methods: the aggregate method or the disaggregate approach (Kavetski et al., 2006a, 2006b; Kuczera et al., 2006; Vrugt et 119 al., 2008; Reichert and Mieleitner, 2009; Renard et al., 2010, 2011). The aggregate method, on which this research is focused, 120 deals with modeling the hydrological model errors, considering the aggregated effects on them (i.e. bias, autocorrelation, 121 heteroscedasticity, etc.), which are produced by all the uncertainty sources. This is made without the need to refer the 122 particular contribution of each uncertainty source to these effects. If we are able to pack all the knowledge about the 123 deviations of the model results from the observations, in an appropriate likelihood function (actually an error model), and the 124 observed data meet the system observability conditions (Gelb, 1974), only then it will be possible a correct hydrological 125 parameter estimation which yields unbiased, accurate and physically meaningful parameter values (Sorooshian and Gupta, 126 1983). This estimation necessarily needs to be a joint inference of error and hydrological models in order to avoid biased parameters (Evin et al., 2014). Moreover, since the modeling of the joint distribution  $p(y, y_c)$  can be equivalent, under 127 certain assumptions which will be exposed in the following section 2.2, to the modeling of the joint distribution of the errors 128 129 and the model predictions  $p(e, y_s)$ , then a correct error model will also yield correct predictive uncertainty estimation. So, 130 the answer to the question posed in the title of this section is that the closer we get to the ideal conditions for a correct error

modeling, which may be a difficult task in hydrology (Smith et al., 2015), the more exact and accurate will be both the parameter estimation and the predictive uncertainty evaluation.

#### **133 2.2 The error model**

The relation between a predictand y and a model prediction  $y_s$  can be defined as the composition of: i) a structural part which directly models the predictand expected dependence from the predictors, and ii) an additive or multiplicative random error  $\mathcal{E}$ . From the probability distribution of this random error, it is possible to derive the conditional probability distribution of the predictand. In fact, both distributions are the same when an additive error model is considered, namely  $p(y-y_s, y_s) = p(e, y_s)$ . When this is the case, we rely on the following relation:

$$y = y_s \left( \left\{ \boldsymbol{\theta}_h, \boldsymbol{\theta}_e \right\}, \tilde{\mathbf{X}}, \tilde{\mathbf{s}}_0 \right) + e$$
(1)

Equation (1) states that a predictand value y is obtained through the sum of a model prediction  $y_s$ , and the error, "residual" or deviation e, which aggregates all sources of uncertainty. Model prediction,  $y_s$ , is expressed as a function of an observed set of "k" predictors (or model inputs)  $\tilde{\mathbf{X}} = \{\tilde{\mathbf{x}}_{1:N}^1, ..., \tilde{\mathbf{x}}_{1:N}^k\}$ , the set of hydrological and error model parameters  $\{\boldsymbol{\theta}_h, \boldsymbol{\theta}_e\}$ , and the initial conditions  $\tilde{\mathbf{s}}_0$ .

The aggregated error e can be decomposed in two components:

$$e = \mu_{e|v_{e}|v_{e}|} + \varepsilon \tag{2}$$

The first one in Eq. (2) is a systematic component,  $\mu_{e|y_s}$ , generally modeled as deterministic, and which could be non-147 constant. The second component  $\mathcal{E}$ , is a random variable with zero mean and whose variance could also be variable. It is 148 very important to note that  $\mu_{e|y_s}$  is an error shifting function, which is only able to represent the expectations of the error 149 conditional distributions (the error conditional bias), namely  $\mu_{e|y_s} = E[e|y_s]$ , when the inferred errors fulfill the Total 150 Expectation Law (TEL), also called the Iterated Expectations Law, or Adam's Law:

$$E\left[e\right] = E\left\{E\left[e|y_s\right]\right\}$$
(3)

where,  $E[\bullet]$  is the expectation operator. In other words, Eq. (3) tells us that the marginal (total) expectation of the error is

equal to the expectation of all error conditional (on  $y_s$ ) expectations.

Besides, Eq. (1) could also be written as:

$$y = E\left[y \mid y_s\right] + \varepsilon \tag{4}$$

where  $E[y|y_s] = y_s + \mu_{e|y_s}$  is the deterministic part of the predictand, and  $\mathcal{E}$  is an additive random error. In the case that a hydrological model would yield unbiased outputs then  $\mu_{e|y_s} = 0$  and so  $E[y|y_s] = y_s$ . It is important to note that we can have a zero-mean error marginal distribution, but this is not equivalent to have hydrological model without bias, since conditional biases  $\mu_{e|y_s} \neq 0$  could auto-compensate yielding a total expectation E[e] = 0. Bias is a non-random deviation in the simulated  $y_s$  value, due mainly to hydrological model structural deficiencies and systematic errors in forcing input data.

In this research, and without loss of generality, the basin outlet discharge is considered as the simulated variable of predictive 163 interest. In fact, this is the general practical situation and we have selected it as the only explicative variable for the bias. In 164 particular, we have assumed a straightforward double linear bias model defined by:

$$\mu_{e|y_s} = \gamma \qquad if \qquad y_s \le y_0$$
$$\mu_{e|y_s} = \gamma + \tau \left(y_s - y_0\right) \quad if \qquad y_s > y_0 \tag{5}$$

where  $\gamma$ ,  $\tau$  and  $y_0$  are error model parameters to be inferred jointly with the hydrological ones (in fact, all error model 167 parameters will be jointly inferred with the hydrological ones). This double function aims at the consideration of two 168 different expected error behaviors distinguishing the low flows (with a constant bias) from the high flows (with a linear bias).

As aforementioned, errors could exhibit a non-constant variance. Following Schoups and Vrugt (2010), Evin et al. (2013,

= 2014) and others, we assume in this paper a linear heteroscedasticity model, where  $y_s$  is the only explicative variable for the

conditional error standard deviation  $\sigma_{e|y_{x}}$ . In this case, we can write:

$$\sigma_{e|y_s} = \alpha + \kappa y_s \tag{6}$$

where  $\alpha$  and  $\kappa$  are error model parameters. It is important to note that function  $\sigma_{e|v}$  represents to the standard deviation

of the error conditional distributions, namely  $\sigma_{e|y_s}^2 = V[e|y_s]$ , only if the inferred errors fulfill the Total Variance Law (TVL), also called the Variance Decomposition Law or Eve's Law:

$$V[e] = E\left\{V[e|y_s]\right\} + V\left\{E[e|y_s]\right\}$$
(7)

where  $V[\bullet]$  is the variance operator. Equation (7) tells us that the marginal (total) variance of the error is equal to the sum of two terms: the first term assesses the expectation of the conditional (on  $y_s$ ) variances of the error, and the second term 178 evaluates the variance of the error conditional (on  $y_s$ ) biases. On the contrary, when TVL is not fulfilled, function  $\sigma_{e|y}$ 179 180 does not represent to the standard deviation of the error conditional distributions, and an error standardization process by 181 using this function would be a simple error scaling, instead of a correct standardization. 182 Having defined the models for the error bias and the error variance, we can also consider the possibility that the random 183 component of the errors,  $\mathcal{E} = e - \mu_{e|v_e}$ , still exhibits serial correlation. Theoretically, the more accurate the bias model is, 184 the smaller the remaining error serial correlation shall be. This dependence (error autocorrelation) can be due to the 185 "memory" effect, caused by the propagation of forcing and structural errors through model storage components (Kavetski et 186 al., 2003). The error autocorrelation can be modeled, as in Schoups and Vrugt (2010), Evin et al. (2013, 2014) and others, 187 using an autoregressive (AR) model. At this stage the modification proposed by Evin et al. (2013) about the methodology 188 applied in Schoups and Vrugt (2010) is considered: that is, errors should be studentized before applying an autoregressive

error model on them. According to Evin et al. (2013), the reason for this change in the original method lies in the mathematical behavior of the autoregressive equations, and is particularly related to the error accumulation properties which

are different in both approaches (for more details see Evin et al. (2013)). So, the standardized errors are defined as:

$$\eta = \sigma_{e|y_s}^{-1} \left( e - \mu_{e|y_s} \right) = \sigma_{e|y_s}^{-1} \varepsilon$$
(8)

- where  $\mathcal{E}$  is the zero-mean additive random error,  $\sigma_{e|y_s}$  is the modeled standard deviation of the error conditioned on the  $y_s$
- value and  $\mu_{e|y_e}$  is the modeled mean of the error (actually its bias) conditioned also on  $y_s$ .

Coming again to the matter of the errors autocorrelation, the equations of pure autoregressive model have the following compact form (Box and Jenkins, 1994):

$$\left(1 - \sum_{i=1}^{p} \phi_i B^i\right) \eta = \phi_p(B) \eta = z$$
(9)

which expresses the *p*-order autoregressive model over the standardized errors  $\eta$ , where  $\phi_1$  are the *i*-order autoregressive coefficients, *B* is the backshift operator  $B^i \eta_i = \eta_{t-i}$  and *Z* are the resulting innovations. Innovations are mutually independent random errors which represent the measurement errors (e.g. in forcing data and discharge measures). They follow a probability distribution with a constant variance  $\sigma_z^2$ . This variance, which should be considered as another (derived) parameter of the error model, can be evaluated according to the following expression (Box and Jenkins, 1994):

$$\sigma_z^2 = \sigma_\eta^2 \left( 1 - \sum_{i=1}^p \rho_i \phi_i \right)$$
(10)

where  $\rho_i$  are the *i*-lagged autocorrelation coefficients of the standardized error series and  $\sigma_{\eta}^2$  is the variance of the standardized errors. In this research we have used an AR(1) model as in Schoups and Vrugt (2010) and Evin et al. (2013, 2014). Since the innovations  $\mathcal{Z}$  are not standardized, they are subjected to a final transformation using their standard deviation:

$$a = \sigma_z^{-1} z \tag{11}$$

where  $\sigma_z$  is obtained from Eq. (10) and a are the final standardized innovations, which is an independent random variable with zero mean and unit variance (namely a standard white noise). Substituting Eq. (8) in Eq. (9) and this in Eq. (11) we obtain the relation between errors and innovations:

$$a = \sigma_z^{-1} \phi_p \left( B \right) \left( \sigma_{e|y_s}^{-1} \left( e - \mu_{e|y_s} \right) \right)$$
(12)

At this point, we should establish the pdf of the standardized innovations obtained in Eq. (12). This research models the standardized innovations using the Skew Exponential Power SEP(0,1,  $\xi$ ,  $\beta$ ) pdf, with two parameters (skewness  $\xi$  and kurtosis  $\beta$ ). The SEP(0,1,  $\xi$ ,  $\beta$ ) pdf, as shown in Schoups and Vrugt (2010), offers great flexibility by avoiding the a priori assumptions about specific forms of the innovations probability distribution. In fact, a SEP(0,1,  $\xi$ ,  $\beta$ ) pdf may adopt a variety of forms, from the Normal to Laplace distribution, as well as it is capable to reproduce asymmetries and heavy tails. The analytical expression of the SEP(0,1,  $\xi$ ,  $\beta$ ) pdf is defined as follows:

$$p(a \mid \xi, \beta) = \frac{2\sigma_{\xi}}{\xi + \xi^{-1}} w_{\beta} \exp\left(-c_{\beta} \left|a_{\xi}\right|^{\frac{2}{1+\beta}}\right)$$
(13)

where

Page 6 of 39

$$a_{\xi} = \xi^{-\operatorname{sign}(\mu_{\xi} + \sigma_{\xi} a)} \left( \mu_{\xi} + \sigma_{\xi} a \right) \tag{14}$$

is a function of the standardized innovations a, and  $\mu_{\xi}, \sigma_{\xi}, c_{\beta}, w_{\beta}$  are a function of skewness  $\xi$  and kurtosis  $\beta$

parameters, as shown in Schoups and Vrugt (2010).

As summary, the vector of error model parameters is  $\boldsymbol{\theta}_e = \{\alpha, \kappa, \gamma, \tau, y_0, \phi_1, \xi, \beta\}$  and  $\boldsymbol{\sigma}_z$  could be considered as a derived parameter. A major question related with the error model parameters must be pointed out. The application of the TLs

on the error bias and variance models reduces the degrees of freedom in the Bayesian inference problem, and consequently,

also reduces the number of independent error model parameters, namely the dimension of the parameter space. This will beclearly exposed in section 4.4.

# 229 2.3 Why and when is imperative the enforcement of the Total Laws

Firstly, we want to emphasize a key concept which was previously exposed in the introduction: predictive uncertainty must 231 be linked with the existence of a joint probability distribution of the predictand observations and the related model 232 predictions. Under the hypothesis of an additive error model, this will be equivalent to considering the joint probability 233 distribution of the predictand errors and the model predictions, as it was made for example in Montanari and Brath (2004), 234 albeit they worked the statistics in the NQT (Kelly and Krzysztofowicz, 1997) transformed space. We are actually modeling 235 the conditional distribution of the error, given the model prediction. In other words, we are modeling indirectly the above 236 mentioned joint probability distribution of errors and model predictions. This means that the error conditional distributions 237 must fulfill the proper restrictions, in order to ensure that all of them make up the full joint distribution. The enforcement of 238 the Total Laws (TLs) exposed in Eqs. (3) and (7) allows us to impose the necessary restrictions in order to achieve that the 239 conditional and marginal distributions of the inferred errors belong to the same bidimensional joint distribution of these 240 errors and the modeled state variable of interest (the simulated streamflow in this research).

An incorrect error model will yield biased hydrological parameters (Sorooshian and Gupta, 1983) and, obviously, an incorrect uncertainty assessment. Two main causes can lead to this situation: i) an incorrect hypotheses about the error conditional distribution features (shape, mean and variance) or about the errors dependence structure; ii) making inferences without enforcing the TLs when we are modeling the error and state variable joint distribution through the definition of its conditional distributions. In order to better understand the TLs importance and implications, Fig. 1 shows what the TLs entail under different hypotheses for the error variance and bias models.