# Peer review of "Bayesian joint inference of hydrological and generalized error models with the enforcement of Total Laws"

_Hydrology and Earth System Sciences, 2017_

## Referee Comment (RC1) · Anonymous Referee #1 · 8 Feb 2017

The manuscript investigates residual error models for the calibration of conceptual hydrological models. The authors rightfully point out that the previous literature has identified failures in the joint calibration of hydrological and error model parameters under particular conditions, eg, for the GL approach (Schoups and Vrugt 2010) and for the WLS-AR1 approach (Evin et al 2013, 2014).

I generally found the study interesting as it explored an important aspect of hydrological model calibration using statistical techniques. The implications of the Total Variance Laws on uncertainty estimation in hydrological models is certainly worthy of research attention and this manuscript does provide some insights towards that aim. There are other interesting results, eg, some of the analyses around Fig 15 are instructive and

visually well presented.

Unfortunately, there appear to be several major flaws in the study:

**1. METHOD SELECTION**

The aim of the study is to address the failure - via unstable/explosive prediction limits - of joint inference approaches. However, this failure has been reported only specifically when inferring the autocorrelation parameter rho - the manuscript notes this on lines 39-40 citing Evin et al 2014, but maybe overlooks that Evin 2014 show that if rho is fixed, there is no instability.

The manuscript then uses the SLS and WLS approaches as a major part of the analysis - even though neither of these methods have an autocorrelation parameter, let alone infer it! So WLS (and SLS), even if they are joint inference methods, do NOT suffer from the instability the study is trying to resolve, and this has already been known from the cited previous studies.

To demonstrate how the TVL approach removes the instability shown by Evin et al 2014, the WLS-AR1 approach from Evin et al should be clearly included in the analysis, which is the error model where the instability actually occurs.

**2. CASE STUDY CATCHMENT**

The catchment used in the manuscript to demonstrate its contribution is the French Broad River from the MOPEX database. This is a very strange choice of catchment for the given research objectives, because it is one of relatively few catchments where pretty much every residual error model has performed well, including the original GL approach of Schoups and Vrugt 2010, and the joint WLS-AR1 scheme of Evin et al 2013, 2014, as can be seen from those papers.

In this respect it should be clear that using such a case study catchment cannot provide supporting evidence of the conclusions.

If the authors wish to demonstrate they have "solved" the problems with the above error models, I think it should be obvious that the case study should *at least* use a catchment where the old models fail in the sense of producing clearly explosive prediction limits (the problem the study is trying to solve) and the new model shows significant improvement (the outcome the study is trying to achieve).

3. GENERAL SUPPORT FOR CONCLUSIONS

I struggle to see empirical evidence in support of the conclusions, which begin with "This paper has addressed the challenging problem of jointly estimate hydrological and error model parameters in a Bayesian framework, trying to solve some of the problems found in previous related researches, as in the second case study of Schoups and Vrugt (2010) as well as in Evin et al. (2014), among others".

As already mentioned above, neither the method nor catchment selection can support such conclusion.

But even if we consider Figure 15 which compares the reliability and resolution of the error models. The WLS error model that the manuscript claims to improve on is clearly amongst the best of the error models under consideration. It clearly has other deficiencies, such as lack of treatment of AR1, but this has already been remedied in the literature by including an AR1 term (Evin 2014).

So I struggle to see how the conclusions can refer to having addressed problems with this error model. There may be improvements related to the treatment of non-Gaussian errors by including skew and kurtosis in the GL error model, but this has already been shown by Schoups and Vrugt in 2010.

4. APPARENT TECHNICAL ERRORS

The discussion on pages 644-657 acknowledges the good performance of WLS to some extent, which helps. However this discussion does not appear in any way anchored to the previous theoretical presentation. To raise these points in the discussion,

there has to be a corresponding background presentation of what is a "bad-posed" problem, how is it problematic and how to detect it. The way the manuscript reads at the moment, the study reached unexpected conclusions (which happens) but instead of re-thinking some of its key premises, it tries to "patch" it in the report using concepts that just haven't been properly introduced at that stage of the presentation.

Unsurprisingly, erroneous, or least confusing, statements appear to be introduced in this "patch". For example, Line 654 states that in the WLS error model, the same hydrological parameter "estimation" is inferred, as well as similar uncertainty bands obtained, using any multiple of (a,k)_ML,TL. Here (a,k) are parameters of the standard deviation of residual errors, sigma=a+k*Q and Q is the streamflow. I really struggle to see how this makes any sense - if we multiply (a,k) by some value c as suggested, sigma will increase by the same factor and the uncertainty bands will inflate accordingly. And the likelihood value will certainly be different. Perhaps this paragraph is missing some major extra detail, or maybe it is a wording issue, or maybe there is some other error/omission in the analysis or calculation, but as it stands it makes no sense. The text states "it can be demonstrated" - please do include the (mathematical) demonstration as it will clarify what you are trying to show.

**5. THE PRESENTATION IS TOO DISORGANIZED**

There is a reason why technical reports have a generally agreed standard set of sections, such as Intro, Theory, Methodology, Results, Discussion, etc. This allows the reader to easily find any details they are interested in. At the moment the specifics of the case study are scattered all across the paper and its very hard to ascertain exactly what experiments were undertaken and why. Please consolidate the presentation and explanation of the methodology in a single section, as this will avoid (likely) confusion on the part of reviewers and readers.

Articulating a more precise set of objectives early on, and justifying how the paper will prove these objectives have been achieved, would also help the reader navigate the

paper.

———

In conclusion, even as I consider the theoretical investigations presented in the manuscript to show promise towards resolving the study objectives, the methodology is clearly inappropriate to demonstrate what the manuscript hopes to achieve. There are other important issues than need to be clarified and improved prior to resubmission.

For this reason I cannot recommend publication of this manuscript in its current form. I encourage the authors to apply their analysis in a thoughtfully designed and presented case study - if convincingly demonstrated, their ideas would provide a worthwhile contribution to the hydrological community.

---

## Referee Comment (RC2) · Anonymous Referee #2 · 21 Feb 2017

The study proposed by Mario R. Hernández-López and Félix Francés follows previous investigations which aim at jointly inferring the parameters of conceptual hydrological models and parameters of residual error models. This is a challenging task, since residual errors exhibit several statistical properties (heteroscedasticity, autocorrelation, asymmetry) which strongly depart from standard assumptions (iid, Gaussianity). The authors try to solve identifiability issues encountered for these parameters using constraints on the variance of these residual errors. More specifically, they decompose this variance by conditioning to the simulated streamflow.

From a theoretical point of view, this work presents a lot of interest since it focuses on global statistical properties of the residual errors and the idea of using constraints in

order to obtain some desired properties is appealing. However, the results shown in the paper do not support their conclusions (see discussion). The following paragraphs discuss numerous points partly addressed or overlooked in the manuscript.

Case study:

The French broad river catchment is a particularly wet catchment with a high annual runoff (800 mm), a high runoff coefficient (0.56), and a very small proportion of low flows. Conceptual hydrological models usually perform well for this type of catchment. As shown in Evin et al. (2013) and Evin et al. (2014), this catchment is atypical in the sense that adequate predicted streamflows (i.e. reliable and precise) are obtained even when the autocorrelation and heteroscedasticity parameters are jointly inferred. In other words, even unstable calibration schemes perform well on this catchment. I really struggle to see why the authors chose a catchment for which calibration issues are not apparent to demonstrate that their methodology solve calibration issues.

I am also puzzled by the choice of the calibration/validation period. First, they apply the hydrological models on a short five-year period (1962-1966) whereas streamflows are available for a much longer period for this catchment (until 1998). Second, they do not apply the split-sample procedure which seems essential to assess the predictive power outside the fitted period.

A major recommendation is thus to:

1. show the results of the calibration proposed in the paper on all the MOPEX catchments, as in Evin et al. (2014), 2. Apply the split-sampling procedure.

If these two requests are fulfilled, a fair comparison with the results shown in Evin et al. (2014) will be possible.

Presentation:

The current presentation of the manuscript is a bit messy. In particular, the main novelties of the paper are presented in several parts. Section 2.1. is a long introduction

to the idea of conditioning the predicted streamflow to the simulated streamflow, which was already presented at lines 62-72. Section 2.2. presents this idea in mathematical terms. Section 2.3. ("Why and when is imperative the enforcement of the total laws") tries to convince the reader that the proposed methodology is essential before the presentation of the results. I would suggest moving this section in the discussion. Finally, Section 4.4. formalizes how this idea can be applied in practice.

The general tone of the presentation gives the impression that all the previous studies for which their methodology was not applied are incorrect. In my opinion, these developments, as all the other related studies, propose calibration schemes which have different desired properties. For example, Evin et al. (2013) show that applying an AR(1) process to standardized errors usually leads to more stable results than applying the AR(1) process to raw errors. I would not say that we 'must' apply the AR(1) process to standardized errors but this approach is preferable since it leads to a more stable calibration schemes with more reliable and more precise predictive streamflows. In the proposed study, as discussed below, the obtained results are not especially impressive, and do not support statements like "The non-fulfillment of the TLs is statistically incorrect". I would appreciate if the authors could let the reader make its own opinion, without using the word 'must' too often ('must' is employed 34 times in the paper).

Interpretation of the results:

- Lines 545-546: "it is expected that the GL++ parameter estimation could be less biased than the corresponding to those classical schemes of inference". Since the 'true' parameters are unknown, a bias cannot be computed and such a statement cannot be verified. I would suggest removing this sentence.

- Lines 562-563: "In relation to the uncertainty assessment, PP-Plots in Fig. 4 show its correctness for both models." I do not understand this interpretation. In Fig. 4 (and also in Fig. 5), we clearly see that GL++ calibration scheme leads to a systematic overprediction of the streamflows, for both hydrological models.

- Lines 615-617: "Furthermore, looking at right panel of the Fig. 9, it is important to realize that the GL++Bias inference for GR4J model is the only inference that exhibits a significant contribution of parameter uncertainty to the total predictive uncertainty. This contribution seems to be underestimated in all the other performed inferences." I strongly disagree with this statement. The parameter uncertainty is related to the complexity of the calibration scheme. For example, for the SLS calibration scheme, there is only one parameter to estimate for the residual error model, which is easily identified. The parameter uncertainty is thus logically small in this case.

- Lines 717-718: "Nevertheless, the most plausible inferred value for $\theta2$ (the closest to zero) corresponds to GL++, the most correct among these three error models." The water balance parameter $\theta2$ in GR4J tends to compensate global under/over-estimations. In the absence of physical explanations, this parameter can thus be different from zero in order to reproduce the global volume of water. In this case, it acts as a 'bias' parameter. The fact that it is close to zero with GL++ is unclear to me, but how can the authors claim that it is the 'most correct' estimate when GL++ leads to a systematic over-estimation of the streamflows? For an unexplained reason, with GL++, $\theta2$ is not able to compensate the excess of water produced by the GR4J model, which is certainly not a desirable feature.

Figure 15 shows that the WLS calibration scheme offers the best combination of resolution and reliability. GL++ calibration scheme leads to overpredicted streamflows and G++bias fails when the CRR hydrological model is applied (see the wide predictive limits in Fig. 9). It seems to also fail when the GR4J model is applied, since G++bias leads to meaningless parameter estimates (unrealistic estimates of $\theta2$ in Fig. 14).

If I understand correctly the conclusion, the authors recommend the application of GL++ or GL++bias, the "fulfillment of the error model hypotheses" being the most important criteria. In my opinion, the reliability and the resolution of the predicted streamflows (second criteria) is by far the most important criteria. From an operational point of view, unreliable or imprecise predictions are useless and indicate that the calibration

scheme is inappropriate.

Parameter identifiability:

The development of more complex calibration schemes is usually difficult due to strong parameter interactions and difficulties in identifying all the parameters. This central issue has been extensively discussed in the literature (see, e.g., Renard et al., 2010) but is overlooked in the manuscript. The only exception is Figure 12, which shows that strong parameter interactions between the slope and the autocorrelation parameters are present with GL++NTL, but not with GL++. However, I suspect that other parameter interactions are present and not shown. For example, I would be curious to see the correlations between $\theta 2$ and the other parameters with the GL++ calibration scheme, in particular with the parameter of the bias model in Eq. (5). That would explain the high values of $\theta 2$ in Fig. 14.

At lines 639-641, the authors claim that "This incorrectness generates problems, mainly related with spurious parameter interactions, affecting the inference results and making them unsuitable and possibly non-robust (Evin et al., 2014). This section will demonstrate that not enforcing the TLs is, at least, one of the most important causes of these problems." To be demonstrated, the authors must show that these parameter interactions are systematically removed, and not only for a couple of parameters.

Bias:

Since hydrological models usually have parameters affecting the water balance (as the $\theta 2$ parameter of the GR4J model), I struggle to see how the parameters of the hydrological models can be jointly inferred with the parameters of a bias model. I suspect that strong parameter interactions lead to the meaningless estimates of $\theta 2$ with GL++bias in Fig. 14. Furthermore, in Table 3, the MAP value of the $\gamma$ parameter is exactly -1. This rounded value could indicate that a lower bound has been set to this parameter and that it cannot be identified with GL++bias.

Kurtosis:

The $\beta$ parameter indicates the kurtosis of the SEP distribution. It is associated to the forth moment and can be interpreted in terms of flatness of the distribution. In this study, as in Schoups and Vrugt (2010), this parameter always hits the lower bound ($\beta$=1) and seems difficult to identify. I would suggest trying alternative calibration schemes without the kurtosis component. Calibration schemes with a Gaussian distribution instead of a SEP distribution in GL++ (without the TLs constraints) corresponds to calibration schemes tested in Evin et al. (2014) and could be interesting to compare to the calibration schemes of the manuscript.

Renard, B., D. Kavetski, G. Kuczera, M. Thyer, and S. W. Franks (2010), Understanding predictive uncertainty in hydrologic modeling: The challenge of identifying input and structural errors, Water Resour. Res., 46, W05521, doi:10.1029/2009WR008328.

---

## Author Comment (AC1) · 23 Feb 2017

Dear Reviewer, dear Editor, please see the responses to the reviewer comments in the attachment.

Yours sincerely, Mario R. Hernández-López, Félix Francés

Please also note the supplement to this comment:
http://www.hydrol-earth-syst-sci-discuss.net/hess-2017-9/hess-2017-9-AC1-supplement.pdf

---

## Referee Comment (RC3) · Anonymous Referee #1 · 26 Feb 2017

Mario R. Hernández-López and Félix Francés

**Anonymous Referee #1**

I am afraid I have to firmly disagree with the majority of these arguments. I will more succinctly summarize the key concerns raised in my original review, most notably the un-substantiated conclusions.

The current manuscript clearly uses the findings of Schoups and Vrugt (2010) and Evin et al (2013, 2014) as a motivation - this is clear from the first line of conclusions and from the introduction, and generally throughout the manuscript. So what are the problems raised by SV10 and E13/E14 that have been resolved in the current manuscript?

The publications by SV10 and E13/E14 empirically found that joint inference (WLS-AR1 scheme) that includes the autocorrelation parameter 'rho' is not always stable - it performs well in catchments such as the French Broad River, but falls apart in many

other catchments such as San Marcos, etc. And the manifestation of the problem is explosive behavior of the prediction limits - an issue of clear concern to a modeller. As an alternative to joint inference, E14 offer a post-processing method, WLS-AR1-PP, to avoid the instability problem and produce well-behaved prediction limits, including in the problematic catchments such as San Marcos.

If the manuscript under review is to claim to resolve the problems with joint inference raised by SV10 and E13/E14, it should clearly show a joint-inference technique that produces well-behaved prediction limits in the problematique catchments, with performance at least as good or better than, eg, the post-processing method WLS-AR1-PP from E14. Otherwise where is the actual improvement???

This request is especially salient in view of the authors response to Comment #2 (top of page 2 of response), where they state that "The cause of this 'non-robustness' (reported by Evin et al 2014) is not the possible interaction among parameters. The cause is a statistical one: the non-fulfillment of Total Laws. All other detected problematic (or not) effects, as the possible interactions, are a consequence of this severe statistical mistake."

There is clearly no evidence shown by the current manuscript in support of this suggestion: Evin et al 2014 reported good performance for the WLS-AR1 scheme in the French catchment (and a few others), and reported non-robustness in catchments such as San Marcos (and a few others). The present manuscript only considered the French Broad River.

Therefore one of the manuscript's central assertions fails on at least three counts:

1) There is no evidence given in the manuscript that fulfilling the TVL avoids non-robustness (this requires an application in the problematic catchments such as San Marcos);

2) If violation of TVL is such a severe mistake, why does it not manifest itself in the

application of the WLS-AR1 scheme in half of the Mopex catchments (eg, in the French Broad), and

3) The WLS-AR1-PP scheme does not satisfy the TVL conditions either, yet does not exhibit non-robustness in *any* of the 12 catchments tested by E14.

The manuscript methodology is just not setup to answer any of these questions, which precludes any empirical support for most of its arguments - most notably, with respect to the practical importance of the TVL conditions.

If I read the authors response to the review correctly (comment #6), they consider the fulfillment of TVL more important than properties such as reliability and precision of the predictive distribution. I am a little puzzled by this perspective - if a statistical scheme that satisfies the TVL conditions produces predictions that are no better than predictions obtained when violating the TVL, according to metrics such reliability and precision (which are the key metrics used in the forecasting community), then it suggests that fullfilling TVL does not bring much practical value. Simply fulfilling the TVL conditions can not be considered a *practical* advance in its own right until it is shown to be beneficial in some *practical* performance aspect. This is especially so if enforcing the TVL conditions sacrifices other performance criteria of already-established value to hydrological forecasters and other modellers.

The other points raised in the response - such as that WLS does not capture the error time dependence - have already been resolved in previous work such as SV10 and E13/E14. The methods recommended in these published studies already incorporate AR1 assumptions and in terms of error time dependence the current manuscript does not offer anything beyond the AR1 treatment.

In summary, I am not at all convinced by the authors response - if they wish to claim to have solved the *problem* with joint inference including 'rho', they should demonstrate their technique on a catchment where existing joint inference techniques fail. The current manuscript simply shows a new, more complex technique that, according

to common performance metrics, performs no better than existing techniques on the French Broad river, which is one of the most well-behaved catchments to appear in the hydrological literature, where the majority of hydrological models and error models already perform very well. I do not believe this constitutes a substantiated contribution at the level of a peer-reviewed publication in the hydrological community.

This is not to discount the *potential* contribution of the authors' methods, just that, as it stands, the case study setup precludes any meaningful claims of practical improvement over existing techniques.

—————————————————————

---

## Author Comment (AC2) · 6 Mar 2017

We thank the reviewer for nicely summarizing the key aspects of our study and pointing out their importance. We have addressed all of his comments point by point and we have tried to respond as short and clear as possible. Moreover, we enormous appreciate the interest shown by reviewer, even in the specific details of our work as figures, tables, etc. We have revised the manuscript based on the suggestions and advice of the reviewer and we have done a big effort in the next replies. We hope that our answers successfully address his concerns and requirements, and the proposed modifications for the revised manuscript will be accepted for publication.

**Reviewer's comment #1**

Case study:

The French broad river catchment is a particularly wet catchment with a high annual runoff (800 mm), a high runoff coefficient (0.56), and a very small proportion of low flows. Conceptual hydrological models usually perform well for this type of catchment. As shown in Evin et al. (2013) and Evin et al. (2014), this catchment is atypical in the sense that adequate predicted streamflows (i.e. reliable and precise) are obtained even when the autocorrelation and heteroscedasticity parameters are jointly inferred. In other words, even unstable calibration schemes perform well on this catchment. I really struggle to see why the authors chose a catchment for which calibration issues are not apparent to demonstrate that their methodology solves calibration issues.

**Reply**

Reviewer is partially right. Previous papers of Evin et al. (2013, 2014) did not show any kind of problems with the FB basin. Reviewer has to realize that Evin et al. consider a Gaussian distribution for the innovations, in all cases. However, a posteriori checks that they perform on their inferences show that Gaussian hypothesis is non-realistic, even for the FB basin. In order to improve this point, our manuscript considers a SEP distribution, which allows the additional possibility of modeling the skewness and the excess of kurtosis for the innovations. Having a more flexible inference (two more parameters) is a key difference: although it may seem insignificant, it is the cause which produces the known inference problems (enlargement of the uncertainty bands, getting spurious parameter interactions and yielding unidentifiable autocorrelation parameter), also on FB basin, except when TLs are considered.

We would suggest to the reviewer the reading of the reply **RC1#5** to the reviewer-1.

**Reviewer's comment #2**

I am also puzzled by the choice of the calibration/validation period. First, they apply the hydrological models on a short five-year period (1962-1966) whereas streamflows are available for a much longer period for this catchment (until 1998). Second, they do not apply the split-sample procedure which seems essential to assess the predictive power outside the fitted period. A major recommendation is thus to: 1. show the results of the

calibration proposed in the paper on all the MOPEX catchments, as in Evin et al. (2014), 2. Apply the split-sampling procedure. If these two requests are fulfilled, a fair comparison with the results shown in Evin et al. (2014) will be possible.

**Reply**

A 5-years calibration period is exactly the same period (in length and range of dates) used in Schoups and Vrugt (2010): same basin, same period, and same CRR model (additionally, we have also used GR4J). About this matter, Schoups and Vrugt (2010) state: "*Experience suggests that 5 years of daily streamflow data contains enough information about the parameters of conceptual rainfall-runoff models…*". Brigode et al. (2013) states that "*there is no clear consensus on the minimum length of calibration period for rainfall-runoff models, which is probably attributable to the specificity of the catchments and models used in those studies. Specifically … Anctil et al. (2004) obtained good GR4J performance with 3 to 5 year calibration periods and Perrin et al. (2007) showed that the calibration of the GR4J … with the equivalent of only 1 year of data can provide acceptable performance*". Actually, in our experience with better models (an also with more parameter involved), calibration period is possible with just 1 year of daily data (Frances et al., 2007).

We will add this information in the revised manuscript.

Regarding the issue of the validation using a split-sample procedure, the topic of our paper is mainly about the hydrological model calibration via a statistical methodology. The paper tries to explain and solve the theoretical pitfall which is committed in a joint inference when TLs are not considered. Therefore, it is only necessary the comparison of the methodology with and without the mistake during calibration. We do not intend to demonstrate if GL++ and GL++Bias error models and the used hydrological models are absolutely the best ones for the analyzed case study: we agree, this would require a thorough validation at different periods to the calibration one. But, this is not the goal of the manuscript and we wanted to be as short as possible.

We will make this clarification in the goals defined in the introduction of the revised manuscript.

**Reviewer's comment #3**

Presentation:

The current presentation of the manuscript is a bit messy. In particular, the main novelties of the paper are presented in several parts. Section 2.1. is a long introduction to the idea of conditioning the predicted streamflow to the simulated streamflow, which was already presented at lines 62-72. Section 2.2. presents this idea in mathematical terms. Section 2.3. ("Why and when is imperative the enforcement of the total laws") tries to convince the reader that the proposed methodology is essential before the presentation of the results. I would suggest moving this section in the discussion. Finally, Section 4.4. formalizes how this idea can be applied in practice.

**Reply**

Regarding the structure of the document, we will change it according to the suggestions of reviewer-1 defined in RC1#9.

The proposed methodology tries to transpose, what occurs with the error marginal and conditional pdf's in a SLS inference, to other more "sophisticated" schemes of inference. We have not coined the Total Laws. We have only understood that these Laws always must (from a theoretical point of view) be fulfilled: by its own (as in SLS) or enforcing them. For this reason, we do not have to wait for results to assert that Total Laws must be theoretically always fulfilled. For this reason, the manuscript introduces the theoretical concept of TLs from the beginning, without waiting for the case study results.

**Reviewer's comment #4**

The general tone of the presentation gives the impression that all the previous studies for which their methodology was not applied are incorrect. In my opinion, these developments, as all the other related studies, propose calibration schemes which have different desired properties. For example, Evin et al. (2013) show that applying an AR(1) process to standardized errors usually leads to more stable results than applying the AR(1) process to raw errors. I would not say that we 'must' apply the AR(1) process to standardized errors but this approach is preferable since it leads to a more stable calibration schemes with more reliable and more precise predictive streamflows. In the proposed study, as discussed below, the obtained results are not especially impressive, and do not support statements like "The non-fulfillment of the TLs is statistically incorrect". I would appreciate if the authors could let the reader make its own opinion, without using the word 'must' too often ('must' is employed 34 times in the paper).

**Reply**

We are very sorry about the impression that the manuscript produces on the reviewer. Our paper brings to light theoretical concepts not used in previous researches. This does not mean that previous papers are useless. Surely our research will be also overcome in the future, but this is not the important thing. Our deep and sincere intention is only making a small but useful contribution to the hydrological community.

Having said that, regarding the sentence "*I would not say that we 'must' apply the AR(1) process to standardized errors but this approach is preferable since it leads to a more stable calibration schemes…*". If reviewer reads the comment **RC1#5**, he will have the arguments to understand the reason why we say "must" in that sentence: from our experience, recommendation of Evin et al. (2013) is necessary, jointly with the application of TLs, to avoid the "feared" and meaningless uncertainty band enlargement, even in the FB basin. Anyway, thanks to the comment of reviewer, we become aware that we have abused of word "must". We will re-read the manuscript for correcting this issue. For now, **Line 398** of original manuscript will be modified as: "*Firstly we have included the recommendation given by Evin et al. (2013). As they mentioned*

*and our experience confirms, errors should be studentized before applying an autoregressive model on them.*"

Regarding the sentence "*I would appreciate if the authors could let the reader make its own opinion, without using the word 'must' too often*". We are very sorry for this abuse of "must". Some of them are related with our belief that TLs always must meet, **from a theoretical point of view**. We have re-read the entire manuscript in order to improve and to smooth the expressions in the revised manuscript. For example, a compromise solution will be using:"must, from a theoretical point of view" and "should" in all other cases.

**Reviewer's comment #5**

Interpretation of the results:

- Lines 545-546:  "it is expected that the GL++ parameter estimation could be less biased than the corresponding to those classical schemes of inference". Since the 'true' parameters are unknown, a bias cannot be computed and such a statement cannot be verified. I would suggest removing this sentence.

**Reply**

We take and share that idea from Schoups and Vrugt, (2010) (among others) and the proper reference is included in our manuscript in the same **Line 546**. They state: "***Violation of SLS assumptions*** *may introduce bias in estimated parameter values and affect parameter and predictive uncertainty [Thyer et al.,2009]*".

They also state: "***Robustness of the GL inference results*** *can be attributed to three factors: (1) by* *accounting for heteroscedasticity* *less weight is given to high flows, making the inference less sensitive to large flow events in different calibration data sets; (2)* *long tails of the Laplace distribution* *allow for a larger number of large errors, which again induces robustness against outliers and random variations in large flow events; (3)* *accounting for autocorrelation* *in the residual errors filters out measurement, model input, and model structural errors, resulting in less biased and more consistent parameter estimates [Vrugt et al., 2005]*"

From those three references (Schoups and Vrugt, (2010); *Vrugt et al., (2005); Thyer et al., (2009)*) and others as Sorooshian and Dracup (1980) or Sorooshian and Gupta (1983), and from our own experience, it can be concluded that: a correct error model which correctly considers those three elements (heteroscedasticity, dependence, non-Gaussianity) should theoretically yield more robust parameters (with less bias) and more robust predictive distributions, than the classical error models.

As we mention in **Lines 547-549**, "…*this is the reason for the poor performance shown by the biased prediction of the hydrological model: the most plausible parameter set, for both hydrological and error models, brings out (in form of a prediction bias) the deficiencies in the hydrological model and/or in the input data*". That is to say, when there are problems with data and/or with the hydrological model conceptualization, there is a tradeoff between getting a good fitting of the hydrological model to the

observations and getting a reliable calibrated parameter set. To get both things is theoretically impossible. This fact is shown by the GL++ error model of our manuscript.

In the revised manuscript, we will add to this paragraph, all mentioned references. Most of them were already included in other parts of the original manuscript. We will also complete the explanations.

**Reviewer's comment #6**

- Lines 562-563: "In relation to the uncertainty assessment, PP-Plots in Fig. 4 show its correctness for both models." I do not understand this interpretation. In Fig. 4 (and also in Fig. 5), we clearly see that GL++ calibration scheme leads to a systematic overprediction of the streamflows, for both hydrological models.

**Reply**

PP-Plots are a useful tool which is able to show us two different but related aspects about the predictive distribution. The first one is related with the bias of the predictive distribution, that is to say, with the shifting of the predictive distribution relative to the observations. Only when the PP-Plot crosses the diagonal line at probability 0.5, the predictive distribution is unbiased; otherwise we will have a systematic Overprediction/Underprediction. The second aspect shown by PP-Plots is about the correctness of the width of the bands, that is to say, the "quality" of the uncertainty estimation. For example, an inverse S-shaped PP-Plot (typically related with SLS error model, as reviewer can see looking at blue solid lines in **Figure 4)** indicates an Overestimation of the predictive uncertainty. On the contrary an S-shaped PP-Plot would indicate an Underestimation of the predictive uncertainty. Reviewer has to realize that these explanations are valid if the axes configuration is as in our manuscript. Some papers have inverted the axes; therefore the explanations will be also inverted. In the manuscript, **Line 487**, we reference the papers in which reviewer can find a much better exposition about this tool.

Therefore, concerning the **Lines 562-563** about GL++, the explanation is as follows. The first aspect to notice is the related with the bias of the predictive distribution: since the black solid line in **Figure 4** does not cross to the diagonal at 0.5 (actually in any point) the predictive distribution is clearly biased. As the PP-Plot is under the diagonal there is an Overprediction: the more the distance of the PP-Plot to the diagonal, the greater the bias of the predictive distribution. Regarding the second aspect, the uncertainty assessment which is related with the amplitude of the bands, as the PP-Plot does not have an S-shape, we can say that the uncertainty estimation is correct.

We will improve these explanations in the revised manuscript.

**Reviewer's comment #7**

- Lines 615-617: "Furthermore, looking at right panel of the Fig. 9, it is important to realize that the GL++Bias inference for GR4J model is the only inference that exhibits a significant contribution of parameter uncertainty to the total predictive uncertainty. This

contribution seems to be underestimated in all the other performed inferences." I strongly disagree with this statement. The parameter uncertainty is related to the complexity of the calibration scheme. For example, for the SLS calibration scheme, there is only one parameter to estimate for the residual error model, which is easily identified. The parameter uncertainty is thus logically small in this case.

**Reply**

In general, it is true that with less number of parameters, the uncertainty is smaller, if the model is correct. We found underestimation of parameter uncertainty using SLS as it was also found by Schoups and Vrugt (2010) and Thyer et al. [2009] among others. This occurs mainly due to fact that SLS neglects all sources of uncertainty different to measurement uncertainty. In hydrological modeling this can lead to an incorrect parameter inference.

What we say in **Lines 615-617** is just a description of the results. GL++Bias is the only inference which exhibits wide yellow bands (predictive uncertainty due to parameters). And the responsible of this widening is the $\theta 2$ parameter of GR4J model. In **Table 3**, the coefficient of variation for $\theta 2$ in GL++ is larger (0.21) than in GL++Bias (0.16). That is to say, $\theta 2$ parameter is more uncertain in GL++ than in GL++Bias. But, we can observe in **Figure 7–right**, how in GL++, parameter $\theta 2$ is not able to propagate its larger uncertainty through the hydrological model: yellow band is hardly appreciable. However in GL++Bias, $\theta 2$ smaller uncertainty produces a wider yellow band. The reason is that, in GL++Bias, the hydrological model is more sensitive to the value of $\theta 2$ than in GL++. In other words, in GL++Bias, the data support strongly the optimal (MAP) value adopted by $\theta 2$, since variations from this optimal are not possible without modifying considerably the output.

The aim of these explanations in the manuscript is merely to make a description of what occurs, and the main comparison is between GL++ and GL++Bias, since both are complex error models. The following **comments #8 to #11** perhaps clarify better the issues about $\theta 2$ parameter.

Of course, we will add these explanations in the revised manuscript.

**Reviewer's comment #8**

- Lines 717-718: "Nevertheless, the most plausible inferred value for $\theta 2$ (the closest to zero) corresponds to GL++, the most correct among these three error models." The water balance parameter $\theta 2$ in GR4J tends to compensate global under/over-estimations. In the absence of physical explanations, this parameter can thus be different from zero in order to reproduce the global volume of water. In this case, it acts as a 'bias' parameter. The fact that it is close to zero with GL++ is unclear to me, but how can the authors claim that it is the 'most correct' estimate when GL++ leads to a systematic over-estimation of the streamflows? For an unexplained reason, with GL++, $\theta 2$ is not able to compensate the excess of water produced by the GR4J model, which is certainly not a desirable feature.

**Reply**

Firstly, θ2 is a hydrological parameter whose definition is groundwater exchange coefficient for the deep (or regional) aquifer of the basin. As reviewer correctly underlines, θ2 is able to compensate (and close) the global water balance. But, for doing that, a deep aquifer should exist in the basin. If this is not the case, θ2 should not compensate anything. Therefore, since any deep aquifer exists in this basin (**Lines 711-712**), θ2 parameter should not do anything except adopting a zero or near-zero value. If we want a parameter which acts "as a bias parameter" we should consider it, but in an explicit way in the error model, as GL++Bias does.

Reviewer is right: GL++ leads to a systematic over-estimation of the streamflows. GL++ is acting as a (very general) diagnostic tool. GL++ says to us: "these are the best results that your model and your data are able to yield; if you want better results improve the data, the model or both."

As we previously mentioned (**reply #5**), GL++ brings to light the tradeoff between getting a good fitting of the hydrological model to the observations and getting a reliable calibrated parameter set. To get both things is impossible when model and/or data have problems.

**Reviewer's comment #9**

Figure 15 shows that the WLS calibration scheme offers the best combination of resolution and reliability. GL++ calibration scheme leads to overpredicted streamflows and G++bias fails when the CRR hydrological model is applied (see the wide predictive limits in Fig. 9). It seems to also fail when the GR4J model is applied, since G++bias leads to meaningless parameter estimates (unrealistic estimates of θ2 in Fig. 14).

**Reply**

Regarding the WLS error model, reviewer can see our replies to reviewer-1 **RC1#3** and **RC1#6**.

We agree, GL++ calibration scheme leads to over-predicted streamflows. In fact as previously mentioned, GL++ is acting as an overall diagnostic tool on the inference: GR4J model plus the used input data are not able to reproduce the observations, as SLS makes us believe.

GL++Bias with CRR does not fail. **Figure 9** shows an "ugly" uncertainty band, but its reliability is quasi-perfect, as the green solid line in **Figure 4** shows. Given that the check of the other error hypotheses is at least acceptable (as **Figure 8** shows), we can conclude that the problem is in CRR model and not in the estimated uncertainty band: CRR yields results with large uncertainty in our case study. For finding a faulty inference, reviewer can see **Figures 7 and 13**, which compare GL++ error model with and without the TLs enforcement, for the two hydrological models: we can affirm that GL++NTL (**Figure 13**) fails.

We agree with reviewer: GL++Bias error model is not the best error model for GR4J, and probably it is not also for CRR: it was not our goal to find the best error model for

the analyzed case study. However, by checking the error hypotheses fulfillment we can affirm that they are far better than SLS and WLS (see **section 5** of original manuscript). Regarding the θ2 parameter, we mention in **Lines 717-718** that the most correct θ2 estimation is the closest to zero (yielded by GL++). As GL++ is showing us that GR4J overpredicts (it needs dewater the basin), therefore a correct bias model would be able to correct this situation. Besides, looking at **Figure 14**, reviewer can see how the application of the bias model, at least moves θ2 parameter from the negative value in GL++ to a positive one. Therefore, it seems plausible that a better bias model be able to move the θ2 value towards zero. But, this is out of the scope of this paper and it will be done in future research.

**Reviewer's comment #10**

If I understand correctly the conclusion, the authors recommend the application of GL++ or GL++bias, the "fulfillment of the error model hypotheses" being the most important criteria. In my opinion, the reliability and the resolution of the predicted streamflows (second criteria) is by far the most important criteria. From an operational point of view, unreliable or imprecise predictions are useless and indicate that the calibration scheme is inappropriate.

**Reply**

As we have mentioned previously, the aim of the paper is not to find "the best" error model. However, GL++ and GL++Bias are far better than classical error models, as we exposed previously. The only thing that we recommend is the enforcement of the Total Laws on inferences which consider jointly the variance and dependence models for the errors. The error models for the variance, the bias and the autocorrelation that we have used are susceptible of being improved, without any kind of doubts: but this is not the aim of the manuscript.

Regarding the issue of how important is the reliability and resolution of the predictive distribution; we want to make an instructive reflection. Having a good reliability or resolution should not be important when the errors exhibit high autocorrelation, which indicates a misspecification of the hydrological model: e.g. this occurs with WLS error model. It is important to remember that, model misspecifications diminish the predictive power of the misspecified models, although apparently, they can exhibit good performance in calibration. The underlying cause of this good performance in calibration is the over-fitting through a "forced value" for parameters, rather than the correctness of the modeling. Kirchner, (2006) states "*[…] to advance the science of hydrology, as opposed to the operational practice of hydrology (that is, to improve our understanding of how hydrologic systems work), we need to know whether we are getting the right answers for the right reasons […] advancing hydrologic science, rather than providing better predictions for operational purposes, although of course one hopes that the former may lead to the latter*".

Therefore, from a **"model calibration point of view"**: if we strive for the error modeling (using error models with an adequate complexity) as we do it for the hydrological modeling, we will obtain benefits as, more reliable hydrological parameter

estimation and also more robust predictive distributions. Therefore, we claim: First: The benefits of using a complex error model only arise by performing a joint inference of hydrological and error model parameters. Second: using a complex error model, with the aim of exploiting its benefits, requires obviously the fulfillment of its hypotheses by the inferred errors. Third: The definition of the error model through the definition of its conditional distributions, as made in our paper and all related ones, requires the check of consistency between the error marginal and these error conditional distributions. When this consistency does not occur by its own, we propose to use the TLs enforcement.

That said, if the priority is related only with the **"operational point of view"**, rather than with trying to make a reliable parameter inference jointly with getting reliable predictive uncertainty estimations, the problem is very different. There also exist other much more efficient methodologies, generally used for this tasks (papers from authors Krzysztofowicz and Todini are good references). These methodologies are based on calibrated models (including black-box models), with any SLS-related calibration method, which feed an uncertainty Post-Processor. We explain this in **Lines 98-132** of the manuscript. Reviewer can also read our reply **RC1#6** to reviewer-1 for more ideas about the relative importance of reliability and resolution of the predictive distribution, in a model calibration context.

**Reviewer's comment #11**

**Parameter identifiability:**

The development of more complex calibration schemes is usually difficult due to strong parameter interactions and difficulties in identifying all the parameters.   This central issue has been extensively discussed in the literature (see, e.g., Renard et al., 2010) but is overlooked in the manuscript. The only exception is Figure 12, which shows that strong parameter interactions between the slope and the autocorrelation parameters are present with GL++NTL, but not with GL++. However, I suspect that other parameter interactions are present and not shown.  For example, I would be curious to see the correlations between θ2 and the other parameters with the GL++ calibration scheme, in particular with the parameter of the bias model in Eq.  (5). That would explain the high values of θ2 in Fig. 14.

**Reply**

Right: this is a very important topic even for us! Unfortunately, with the aim of not lengthening the article too much we only presented the result and figure more representative of the main objective of the paper. **Figure 12**, because this figure represents a faulty inference of parameter phi, but only when TLs are neglected. As we explain in **RC1#5**, "*Besides (and the most important), it can be observed the extremely high inferred MAP value for phi (about 0.99) when TLs are neglected. In this case the posterior distribution of phi shows extreme asymmetry, with the mode at the value of one, the upper bound value for autocorrelation parameter. This problem was also reported in Scharnagl et al. (2015) for their Likelihood2. From our point of view, this is a synonym of having a non-identifiable distribution for the autocorrelation parameter,*

*since for phi→1, the AR(1) process becomes nonstationary, as explained in Box et al. (1994)."* Other interactions among parameters could be possible, but they have nothing to do with instability of the inference.

Anyway, we will try to calm the curiosity of reviewer by showing three interesting figures about inferences with the GR4J model (not shown in original manuscript). The two first figures are about the most significant parameter interactions on GL++ and GL++NTL inferences. Top figure is from GL++ and the bottom one from GL++NTL (that is to say, GL++ without TLs). The main points to remark on top figure are:

1- GL++ does not show high correlation among hydrological parameters
2- The highest correlations are between θ1-kappa and θ1-phi. However, all parameters are perfectly identifiable, and this inference does not suffer the enlargement of the uncertainty bands.
3- Correlations of θ2 are moderate or low
4- Correlation between kappa-phi is moderate (0.64)

[Figure]

[Figure]

The main points to remark on bottom figure, about GL++NTL are:

1- GL++NTL shows a significant increment in correlations among hydrological parameters θ1-θ2 (0.39 vs 0.04) and θ1-θ3 (0.80 vs 0.46). Why this undesirable increment in correlations between hydrological parameters? They should not to be so strongly correlated, as GL++ shows. These increments in the hydrological parameter covariance matrix (Jakeman and Hornberger ,1993), when TLs are neglected, indicate that hydrological parameters catch less information from data than they do when inference considers TLs. Hence, from this point of view, GL++NTL inference is less efficient than GL++. This seems to be another more symptom of the theoretical pitfall.

2- Correlations of θ2 are moderate but larger than in GL++, for θ1-θ2 and θ2-kappa

3- θ2 MAP value deserves special attention, because it takes the highest absolute value for all inferences: θ2=-4.92. If we accept that θ2=0 should be the correct value, NTL inference yields the farthest value from the correct, even farther than SLS (θ2=-0.71) or WLS (θ2=-1.37).

4- Correlation kappa-phi is the highest (0.89). However, phi is identifiable with GR4J, differently to what occurs with CRR model (see **Figure 12** in original manuscript).

Therefore, with GR4J there are several cases of parameter interactions. The most harmful, from our point of view, are among hydrological parameters: but these occur only in GL++NTL inference. With CRR we found also the problem of a parameter phi non-identifiable (**Figure 12** in original manuscript).

Finally, we will show the figure of correlations among parameters, corresponding to GL++Bias (also with GR4J).

[Figure]

The main points to remark on figure, about GL++Bias are:

1- Hydrological parameters do not exhibit high correlations (as in GL++)
2- All parameters are identifiable, including "gamma" for the bias model (this replies to the **comment #13**)
3- Reviewer is very right. The highest correlations in matrix correspond to those between θ2 parameter and variance and bias error parameters: θ2-kappa (0.96), θ2-gamma (0.95) and θ2-tau (0.91).
   But, where is the problem? We previously explained (see **reply #8**) that the aim of the bias model, in our case study, is to explicitly recognize the necessity of dewatering the basin. We also mentioned that θ2 is not the parameter which has to do this task. Therefore the error model is complying with its function of trying to substitute to the θ2 parameter in the dewatering task. They are so correlated because they are doing the same function: trying to manage the water excess in the basin. But, in our case, the (erroneous) bias model is strongly dewatering the basin; therefore θ2 becomes positive (to bringing water from… some "virtual" aquifer) in order to compensate the excess of drainage by bias model. In short: 1- a correct bias model is necessary to allow θ2 gets a zero-value; 2- our bias model is not the proper one.

For this comment and the two next, we will add these figures and comments as annex in the revised manuscript, if editor allow us, with the corresponding condensed paragraphs in the main text.

**Reviewer's comment #12**

At lines 639-641, the authors claim that "This incorrectness generates problems, mainly related with spurious parameter interactions, affecting the inference results and making them unsuitable and possibly non-robust (Evin et al., 2014). This section will demonstrate that not enforcing the TLs is, at least, one of the most important causes of these problems." To be demonstrated, the authors must show that these parameter interactions are systematically removed, and not only for a couple of parameters.

**Reply**

With previous reply to **comment #11** for GR4J and with **Figure 12** for CRR, we demonstrate how the actual problems with parameter interactions only arise in the inferences without Total Laws. The main detected problems are: 1- With GR4J the spurious interactions among hydrological parameters and 2- With CRR, the non-identifiable autocorrelation parameter. We are not able to detect, in our case study, another more important effects related with parameter interactions. These problems appear when TLs are neglected and they do not when we enforce TLs. But see our long previous reply!

**Reviewer's comment #13**

Bias:

Since hydrological models usually have parameters affecting the water balance (as the $\theta 2$ parameter of the GR4J model), I struggle to see how the parameters of the hydrological models can be jointly inferred with the parameters of a bias model. I suspect that strong parameter interactions lead to the meaningless estimates of $\theta 2$ with GL++bias in Fig. 14. Furthermore, in Table 3, the MAP value of the $\gamma$ parameter is exactly -1. This rounded value could indicate that a lower bound has been set to this parameter and that it cannot be identified with GL++bias.

**Reply**

See our previous reply to **comment #11**

**Reviewer's comment #14**

Kurtosis:

The $\beta$ parameter indicates the kurtosis of the SEP distribution. It is associated to the forth moment and can be interpreted in terms of flatness of the distribution. In this study, as in Schoups and Vrugt (2010), this parameter always hits the lower (sic) bound ($\beta=1$) and seems difficult to identify. I would suggest trying alternative calibration schemes without the kurtosis component. Calibration schemes with a Gaussian distribution instead of a SEP distribution in GL++ (without the TLs constraints) corresponds to calibration schemes tested in Evin et al. (2014) and could be interesting to compare to the calibration schemes of the manuscript.

**Reply**

We consider the SEP distribution of Schoups and Vrugt (2010), because is more general than Gaussian, but if the inference it requires, would also allow Gaussianity; therefore we have two more error parameters (skewness and kurtosis). As it is shown in Evin et al. (2013, 2014) results, these two additional statistical properties are necessary to model errors more correctly.

We disagree with the reviewer's affirmation "*this parameter always hits the higher bound (β=1) and seems difficult to identify*" The value of "1" is a valid and identifiable one, since it is which yields the maximum kurtosis. Moreover, our GL++Bias inference exhibits (see **Figure 8**) an excess of kurtosis that even SEP is not able to reproduce. Perhaps the Skewed Student distribution, as the used by Scharnagl et al. (2015) would have a better behavior than SEP.

Of course, it would be interesting making other comparisons as with the Skewed Student distribution, using more hydrological models, experiments on more basins and also improving the error variance and dependence models. But we consider it is out of the scope of this manuscript and would enlarge unnecessarily the paper. More research is on-going for future papers.

**References**

Anctil, F., Perrin, C., Andréassian, V., 2004. Impact of the length of observed records on the performance of ANN and of conceptual parsimonious rainfall-runoff forecasting models. Environ. Model. Software 19 (4), 357–368. http:// dx.doi.org/10.1016/S1364-8152(03)00135-X.

Brigode, P., Oudin, L. and Perrin, C.: Hydrological model parameter instability: A source of additional uncertainty in estimating the hydrological impacts of climate change?, J. Hydrol., 476, 410–425, doi:10.1016/j.jhydrol.2012.11.012, 2013.

Frances, F., Velez, J. I. and Velez, J. J.: Split-parameter structure for the automatic calibration of distributed hydrological models, J. Hydrol., 332(1–2), 226–240, doi:10.1016/j.jhydrol.2006.06.032, 2007.

Jakeman, A. J. and Hornberger, G. M.: How Much Complexity Is Warranted in a Rainfall-Runoff Model? are good predictors of streamflow and, Water Resour. Res., 29(8), 2637–2649, doi:10.1029/93WR00877, 1993.

Kirchner, J. W.: Getting the right answers for the right reasons: Linking measurements, analyses, and models to advance the science of hydrology, Water Resour. Res., 42(3), n/a--n/a, doi:10.1029/2005WR004362, 2006.

Perrin, C., Oudin, L., Andreassian, V., Rojas-Serna, C., Michel, C., Mathevet, T., 2007. Impact of limited streamflow data on the efficiency and the parameters of rainfall-runoff models. Hydrol. Sci. J. 52 (1), 131. http://dx.doi.org/10.1623/ hysj.52.1.131.

Vrugt, J. A., B. A. Robinson, and V. V. Vesselinov (2005), Improved inverse modeling of flow and transport in subsurface media: Combined parameter and state estimation, Geophys. Res. Lett., 32, L18408, doi:10.1029/2005GL023940

---

## Author Comment (AC3) · 8 Mar 2017

The first thing we want to mention is that thanks to the reviewers' comments, we have become aware of two general aspects must be improved in the final manuscript. The first aspect is the necessity of smoothing the general tone of the manuscript, since it is transmitting an impression, which is far from our sincere intention. Secondly, it seems we have not explained properly the main target of the paper: the aim was not to find the best error model (neither the best hydrological model) which yields the best performance for the FB basin (see our **reply RC1#6** or our **reply RC2#10** to reviewer-2 for more details, among others, those concerning the metrics for reliability and resolution). The objective of this paper is to construct the full general additive error model developed in Schoups and Vrugt (2010) with the recommendations of Evin et al. (2013) within a "strict" Bayesian joint inference framework. We want to be "strict" following the reflection of Todini and Mantovan (2007): "*Statistical scientists will have a very low regard for the hydrological sciences if we, the hydrologists, pretend to use statistical techniques, but then deliver theoretically incorrect answers and results*". Of course the WLS-AR1-PP and other post-processing solutions can work, but they eliminate potential positive interactions between the two models (hydrological and error) during the parameter inference process and, for this reason, we have not considered them in our research.

Therefore, our initial hypothesis was that, by "merging" both mentioned approaches, we could perform Bayesian joint inference on our models. However, after the construction of the Bayesian joint inference framework, we saw that problems remained. This major setback led us to thinking about the basics of error modeling. We understood the mistake looking at the most basic error model (**Figure 1a** of original manuscript), which underlies to the simple Least Squares method (SLS). The joint inference, to be statistically correct, should assume the existence of the joint probability distribution of the variable to be predicted and its deviation from its observation (the error). Consequently, the relationship between the marginal and conditional distributions of this joint distribution must be taken into account in the inference process. The Total Laws define this relationship, resulting in a reduction of the degrees of freedom in the inference problem. We think nobody questioned these issues in the past, perhaps because with SLS error model it was not needed, since SLS meets TLs by its own hypotheses. But with more complex error models, we should take up those old Laws again.

Once we put on board the Total Laws, we wanted to highlight in one case study the problems that can arise without their application, some of them found in previous researches. With this objective in mind, we think we do not need to unnecessarily enlarge the paper with more case studies, because with FB basin we have already found the following spurious problems:

1- Meaningless enlargement of the uncertainty bands with both CRR and GR4J hydrological models; this problem was also found by Schoups and Vrugt (2010), Evin et al. (2013, 2014) and Scharnagl et al. (2015)
2- Non-identifiable autocorrelation parameter with CRR model; this problem was also found by Schoups and Vrugt (2010) in their second case study, and by Scharnagl et al. (2015) with their Likelihood2.
3- Very high, but identifiable, value for the autocorrelation parameter, with GR4J model; Evin et al. (2013, 2014) also reported this feature. In our opinion and in

some cases, the non-suitability of the AR(1) model could also overlap with the spurious effect.

4- Spurious correlation among hydrological parameters (not shown in manuscript, but shown for GR4J model, in **reply RC2#11**). In our knowledge, none of previous publications has reported this issue.

Our research shows for a limited number of examples how these spurious effects on FB basin appears, when TLs are neglected in a strict joint inference. We have also shown how these effects disappear when TLs are enforced. The magnitude and type of these spurious problems depends on the error and hydrological models, as our manuscript shows. Of course, the basin is also a factor but we do not want to enlarge unnecessarily the paper with a large number of examples. Our paper makes a negative empirical demonstration and actually one fail is enough!

**There is not a biunivocal relationship of the kind: "violation of TLs ⇔ inference shows spurious problems"**. In "Conclusions" of the revised manuscript we will highlight this possible misunderstanding. For example, WLS inference (2 free error parameters) does not exhibit any problem when TLs are neglected. Similarly, Schoups and Vrugt (2010) in their first case study inferred the SEP distribution but without considering skewness parameter (4 free error parameters) and they found no problems. Probably, the problems also do not appear for FB basin, in joint inferences which consider neither skewness nor kurtosis parameters, as in Evin et al. (2014) (3 free error parameters). But we found that including the skewness parameter in the joint inference of the first case study of Schoups and Vrugt (2010) (5 free error parameters), the MCMC inference does not converge. By only applying TLs (this case is not shown in original manuscript) or only applying recommendation of Evin et al. (2013) (this latter case is our **GL++NTL**) inference easily converges, but with the spurious enlargement of the uncertainty band. It is necessary the joint application of both measures to allow the convergence and to avoid spurious effects; this is made in our **GL++** error model.

Therefore, it seems that one pattern emerges: the greater or lesser complexity of the inferred error structure (in our case represented by a bivariate pdf) seems to predetermine the occurrence (or not) of spurious problems during the strict joint inference, when parameters are not properly constrained. FB basin is an "easy" basin, therefore its error structure is not complex: until 4 error parameters are admitted without constrictions and without showing spurious problems. Basins which are more difficult to model yield a more complex error structure: in these cases, the bivariate distribution modeling would admit less than 4 free error model parameters.

TLs could be understood as one of those possible parameter restrictions to avoid the spurious problems during the strict joint inference. However, in our opinion, TLs enforcement is more than an ad-hoc method to restrict the error model parameters. Meeting TLs is, theoretically, a statistical requirement which eventually produces the convenient error parameter restriction. TLs are fulfilled in classical inferences (with SLS method) and we understand that these laws are perfectly transposable to any inference which involves to the inferred variables and its errors.

**References**

Todini, E. and Mantovan, P.: Comment on:'On undermining the science?'by Keith Beven, Hydrol. Process., 1638(January), 1633–1638, doi:10.1002/hyp.6670, 2007.

---

## Referee Comment (RC4) · Anonymous Referee #1 · 11 Mar 2017

I appreciate the effort the authors are taking to clarify their message of the importance of satisfying Total Variance Laws (TVL). But the point is that the case study methodology - most notably, the use of a single catchment, especially one as well-behaved as the French Broad River - is not appropriate to demonstrate and support a research question as broad as "solving problems with joint inference including AR1 components". And the results happen to clearly contradict the proposition that satisfying TVL is "essential" from a practical predictions perspective.

For example, at bottom of page 1 of the last response, the authors state that the French Broad River (FB) case study demonstrates problems such as "meaningless enlargement of the uncertainty bands" and cite Schoups and Vrugt (2010), Evin et al (2013,

2014). In fact these studies report precisely the opposite - joint inference, both with WLS-AR1 and the Skewed Exponential Power error models, performs very well in the FB catchment, with tight and reliable prediction limits. This is achieved without imposing any explicit TVL constraints. The problems reported in previous studies occurred in a different subset of catchments (eg, San Marcos, etc).

So, to be blunt, the French Broad river case study results (and the prior results of Schoups and Vrugt and Evin et al) contradict the manuscript proposition that TVL constraints are essential to obtain good quality (reliable and precise) predictions. This is an interesting conclusion worth reporting, which would require a complete reversal of the current manuscript narrative. I am sure its not the first or last time that statistical approximations produce adequate results (the converse is also true!). It may well be that TVL constraints become relevant in catchments other than the FB river case study (eg, in San Marcos), but obviously this requires to be demonstrated empirically through an appropriately designed case study, including multiple error models, etc.

I would also note that using a single catchment and hydrological model to suggest that methods such as Weighted Least Squares (WLS), which are well-established not just in hydrology but in science in general, represent a "severe statistical mistake", strikes me as patently insufficient. To do so would require far stronger and broader empirical evidence.

I hope this feedback is helpful to the authors as they continue to explore this important research question.

---

## Short Comment (SC1) · 18 Mar 2017

In this manuscript, the authors model jointly the hydrological and error model parameters in a Bayesian framework. I am particularly interested in the mathematical theory of the model. Therefore, I took a closer look at the 2nd and 3rd section. In general trying to model the errors using varying parameters, which depend on the level of the hydrological model prediction is interesting. Furthermore, the derivations are far from trivial and the theoretical results are useful.

It seems that the authors first try to form the errors. As a result, some parameters of the model must be expressed as functions of the other parameters (e.g. implementing the law of total expectation and the law of total variance), with a simultaneous reduction

of the parameter space. Here I think that the title of the manuscript is misleading, since satisfying the two laws must be mandatory. Therefore, the term "enforcement" seems unnecessary. I suggest that less emphasis is given to the enforcement and more to the investigation of various forms of the errors because the latter is the cause for further examining the two laws, while the former is the consequence.

Regarding the mathematical part of the manuscript, I recommend that the authors separate the notation for parameters and variables, to ease understanding of the framework and help the reviewing process. Please see the supplement for more details.

Lastly, I do not understand why we should sample from eq. (19) of the manuscript rather than using the distribution of line 335. This would seem the straightforward approach, considering eq. (18) of the manuscript. Once these issues are addressed, I believe the paper would be a useful contribution.

References

Hemelrijk J (1966) Underlining random variables. Statistica Neerlandica 20(1):1–7. doi:10.1111/j.1467-9574.1966.tb00488.x

Please also note the supplement to this comment:
http://www.hydrol-earth-syst-sci-discuss.net/hess-2017-9/hess-2017-9-SC1-supplement.pdf

**Supplement:**

**Supplement**

Eq. (3) should be written as $E[\underline{e}] = E[E[\underline{e}|\underline{y}_s]]$, while eq. (4) should be written as $\underline{y} = E[\underline{y}|y_s] + \underline{\varepsilon}$. Here we underlined random variables (Hemelrijk 1966).

Secondly from eqs. (1) and (2)

$$\underline{y} := \underline{y}_s + \underline{e} \tag{1}$$

$$\underline{e} := E[\underline{e}|y_s] + \underline{\varepsilon} \tag{2}$$

we obtain

$$\underline{y} = \underline{y}_s + E[\underline{e}|y_s] + \underline{\varepsilon} \Rightarrow \tag{3}$$

$$\underline{y} = \underline{y}_s + E[\underline{y} - \underline{y}_s|y_s] + \underline{\varepsilon} \Rightarrow \tag{4}$$

$$\underline{y} = E[\underline{y}|y_s] + \underline{\varepsilon} + \underline{y}_s - y_s \tag{5}$$

which is not identical to eq. (4) of the manuscript.

Furthermore, from eq. (2) of the manuscript we obtain

$$E[\underline{\varepsilon}] = E[\underline{e}] - E[E[\underline{e}|y_s]] \Rightarrow \tag{6}$$

$$E[\underline{\varepsilon}] = E[\underline{e}] - E[\underline{e}|y_s] \Rightarrow \tag{7}$$

$$E[\underline{\varepsilon}] \neq 0 \tag{8}$$

Therefore, $n$ of eq. (8) of the manuscript must be checked whether it is a standard variable. On the other hand, if we use $\underline{y}_s$, instead of $y_s$ then eq. (4) of the manuscript is confirmed and $E[\underline{\varepsilon}] = 0$.

---

## Author Comment (AC4) · 20 Mar 2017

We thank the colleague Hristos Tyralis for his comments and for having pointed out the importance of this research. We enormous appreciate his dedication. We hope that our answers successfully address his concerns and requirements.

**Comment #1**

In this manuscript, the authors model jointly the hydrological and error model parameters in a Bayesian framework. I am particularly interested in the mathematical theory of the model. Therefore, I took a closer look at the 2nd and 3rd section. In general trying to model the errors using varying parameters, which depend on the level of the hydrological model prediction is interesting. Furthermore, the derivations are far from trivial and the theoretical results are useful.

It seems that the authors first try to form the errors. As a result, some parameters of the model must be expressed as functions of the other parameters (e.g. implementing the law of total expectation and the law of total variance), with a simultaneous reduction of the parameter space. Here I think that the title of the manuscript is misleading, since satisfying the two laws must be mandatory. Therefore, the term "enforcement" seems unnecessary. I suggest that less emphasis is given to the enforcement and more to the investigation of various forms of the errors because the latter is the cause for further examining the two laws, while the former is the consequence.

**Reply**

Firstly, we agree with Dr. Tyralis: "*satisfying the two laws must be mandatory*". We also agree with him about the need of making research on the different error structures originated by the different sources of uncertainty, in hydrological modeling. However, the lumped via of error modeling which we are using makes difficult this task. In any case, our original target was making Bayesian inference of hydrological models rather than making research on errors. But we came across unexpected problems which forced us to change the problem to solve. The fact is that modeling the errors as in Schoups and Vrugt (2010), Evin et al. (2013, 2014), Scharnagl et al. (2015) or ourselves requires the enforcement of Total Laws because if not, they are not fulfilled. We will explain again why is this so, since it seems it has not been well understood with the original manuscript.

Commonly, the error modeling has been performed by using a likelihood function assuming a multivariate Gaussian distribution for the errors, with the definition of its covariance matrix, as for example in Del Giudice et al. (2013) or Reichert and Schuwirth (2012) among others: by using this approach we are already considering implicitly the relation between the marginal and conditional distributions of the multivariate Gaussian. However, Schoups and Vrugt (2010) introduced a different innovative and generalized approach for error modeling, which was also followed by Evin et al. (2013, 2014) and Scharnagl et al. (2015) among others. This more flexible approach is based on the modeling of the statistical features of the error conditional distributions: its variance, bias, kurtosis and skewness. This features are modeled each separately: we can hypothesize different functional expressions for each statistical feature. But one detail was neglected in this approach: we are only modeling the error

conditional distributions, overlooking that they are conditionals within a multivariate distribution. **How are we sure that these conditional distributions are part of a multivariate distribution? How are we introducing in the inference framework the existence of this multivariate distribution?**

Explained with equations and following the notation proposed by Dr. Tyralis. We want to model the joint distribution of a simulated variable and its error, $p(\underline{e}, \underline{y}_s)$ for making inference. But instead, we are modeling the conditionals, $p(\underline{e} \mid \underline{y}_s)$, since this way to proceed can provide us more flexibility in the error modeling. Relation between them is established by the conditional probabilities law, as it follows:

$$p(\underline{e}, \underline{y}_s) = p(\underline{e} \mid \underline{y}_s) p(\underline{y}_s) \tag{1}$$

where $p(\underline{y}_s)$ is the marginal probability distribution of $\underline{y}_s$. On the other hand, the marginal distribution of the errors can be obtained by integrating this joint distribution as follows:

$$p(\underline{e}) = \int_{\underline{y}_s} p(\underline{e}, \underline{y}_s) d\underline{y}_s = \int_{\underline{y}_s} p(\underline{e} \mid \underline{y}_s) p(\underline{y}_s) d\underline{y}_s \tag{2}$$

But, it also holds that:

$$E\left[ p(\underline{e} \mid \underline{y}_s) \right] = \int_{\underline{y}_s} p(\underline{e} \mid \underline{y}_s) p(\underline{y}_s) d\underline{y}_s \tag{3}$$

Therefore, from equations (2) and (3), we have:

$$p(\underline{e}) = E_{\underline{y}_s}\left[ p(\underline{e} \mid \underline{y}_s) \right] \tag{4}$$

This is a **relationship between error marginal and conditional distributions which must always be maintained**. But by only defining the statistical features of $p(\underline{e} \mid \underline{y}_s)$ is not enough to ensure the fulfillment of this relationship.

Moreover, equality of Eq. (4) for distributions is also applicable for their expectations, hence we obtain the **Total Expectation Law (TEL)**:

$$E\left[\underline{e}\right] = E_{\underline{y}_s}\left[ E\left[\underline{e} \mid \underline{y}_s\right] \right] \tag{5}$$

On the other hand, it can be demonstrated that Total Variance Law (TVL) is based on TEL. The steps of this demonstration are outlined in what follows. The marginal variance of the errors can be expressed as $V\left[\underline{e}\right] = E\left[\underline{e}^2\right] - E^2\left[\underline{e}\right]$. By considering TEL on the two right terms of this equation, we can write:

$$V\left[\underline{e}\right] = E_{\underline{y}_s}\left[ E\left[\underline{e}^2 \mid \underline{y}_s\right] \right] - E_{\underline{y}_s}^2\left[ E\left[\underline{e} \mid \underline{y}_s\right] \right] \tag{6}$$

By expanding term inside the outer brackets, on the first right term of Eq. (6), we get:

$$V\left[\underline{e}\right]= \underset{\underline{y}_s}{E}\left[V\left[\underline{e}\mid \underline{y}_s\right]+E^2\left[\underline{e}\mid \underline{y}_s\right]\right]- \underset{\underline{y}_s}{E^2}\left[E\left[\underline{e}\mid \underline{y}_s\right]\right] \qquad (7)$$

And by rearranging Eq. (7) and considering again the relation $V\left[\cdot\right]= E\left[\cdot^2\right]-E^2\left[\cdot\right]$, we reach the final **expression of TVL**:

$$V\left[\underline{e}\right]= \underset{\underline{y}_s}{E}\left[V\left[\underline{e}\mid \underline{y}_s\right]\right]- \underset{\underline{y}_s}{V}\left[E\left[\underline{e}\mid \underline{y}_s\right]\right] \qquad (8)$$

Consequently, we have demonstrated from where TLs (TEL and TVL) arise. There is a relationship between the error marginal and conditional distributions, summarized by TLs, which is not considered by only modeling the error conditional variance or bias. Therefore, we must also establish (enforce) this relationship in an explicit way. We think nobody questioned these issues in the past: 1) with SLS error model it was not needed, since SLS meets TLs by its own hypotheses; 2) by using a multivariate Gaussian likelihood we are directly defining a joint distribution rather than modeling the conditional ones, so more considerations are not needed.

These are the reasons why we make a point in term "enforce" and we have included it in the Title.

**Comment #2**

Regarding the mathematical part of the manuscript, I recommend that the authors separate the notation for parameters and variables, to ease understanding of the framework and help the reviewing process. Please see the supplement for more details.

**Reply**

In case that manuscript be accepted, we will study which change in the notation could be more suitable, for the sake of both simplicity and better understanding.

**Comment #3**

Lastly, I do not understand why we should sample from eq. (19) of the manuscript rather than using the distribution of line 335. This would seem the straightforward approach, considering eq. (18) of the manuscript.

**Reply**

Dr. Tyralis is right. The Eq. (18) is the straightforward approach to obtain the predictive uncertainty. This approach, followed by many authors (e.g. Krzysztofowicz and Todini are good references), is related with the "operational point of view" of hydrology. But our research falls within the model calibration context. Therefore, we try to make a reliable parameter inference jointly with getting also reliable predictive uncertainty estimations. This problem is very different to the former one. We explain this in **Lines**

**98-132** of the manuscript. The cornerstone of our approach is the error model. Once we have the error model, jointly inferred with the hydrological one, we have got, theoretically, well-estimated error and hydrological parameters. Therefore, by using Eq. (19) repeatedly, with several samples of the innovation distribution (SEP in our case) and several samples of the error and hydrological parameters, we can obtain samples from the predictive distribution expressed as in Eq. (18). It is important to point out (see **Lines 337-342**) that we want to obtain an approximation to the distribution of Eq. (18) and not to the distribution indicated in **Line 335**: this later is still conditional on parameters! It is also important to remark that the predictive distribution in Eq. (18) is conditional on observations, forcing inputs and initial conditions, all of them supposed known and certain quantities.

**References**

Del Giudice, D., Honti, M., Scheidegger, A., Albert, C., Reichert, P. and Rieckermann, J.: Improving uncertainty estimation in urban hydrological modeling by statistically describing bias, Hydrol. Earth Syst. Sci., 17(10), 4209–4225, doi:10.5194/hess-17-4209-2013, 2013.

Evin, G., Kavetski, D., Thyer, M. and Kuczera, G.: Pitfalls and improvements in the joint inference of heteroscedasticity and autocorrelation in hydrological model calibration, Water Resour. Res., 49(7), 4518–4524, doi:10.1002/wrcr.20284, 2013.

Evin, G., Thyer, M., Kavetski, D., McInerney, D. and Kuczera, G.: Comparison of joint versus postprocessor approaches for hydrological uncertainty estimation accounting for error autocorrelation and heteroscedasticity, Water Resour. Res., 50(3), 2350–2375, doi:10.1002/2013WR014185, 2014.

Reichert, P. and Schuwirth, N.: Linking statistical bias description to multiobjective model calibration, Water Resour. Res., 48(9), n/a--n/a, doi:10.1029/2011WR011391, 2012.

Scharnagl, B., Iden, S. C., Durner, W., Vereecken, H. and Herbst, M.: Inverse modelling of in situ soil water dynamics: accounting for heteroscedastic, autocorrelated, and non-Gaussian distributed residuals, Hydrol. Earth Syst. Sci. Discuss., 12(2), 2155–2199, doi:10.5194/hessd-12-2155-2015, 2015.

Schoups, G. and Vrugt, J. A.: A formal likelihood function for parameter and predictive inference of hydrologic models with correlated, heteroscedastic, and non-Gaussian errors, Water Resour. Res., 46(10), 1–17, doi:10.1029/2009WR008933, 2010.

---

## Author Comment (AC5) · 3 Apr 2017

First of all, we would want to thank the two anonymous referees and Dr. Tyralis for their suggestions, comments and questions. We are sure they have been useful to improve the manuscript.

We are aware that changes are required in the manuscript, as we pointed out in our replies. However, referees highlighted the good research quality of the manuscript and they found it relevant. Significant number of downloads of the manuscript, from HESS web page, is also a good indicator that our research generates interest in the community. It is always a pleasure to hear that your work is receiving attention and interest and is providing new insights. Now, we are going to summarize the main raised research comments and our replies.

**Firstly, we want to clarify the main target of the paper.** Our research has always been within a Bayesian calibration framework and not on "operational hydrology". This means that, obtaining the best possible reliable parameter set is mandatory for us, hence we only consider the **joint inference** option, namely, inferring all error model parameter jointly with all hydrological ones. Achieving this parameter reliability is possible through an error model which acceptably fulfills all its hypotheses. Moreover, with the compliance of all errors' hypotheses, namely having a reliable error model, we are also able of reliably assessing the hydrological model predictive uncertainty, since the predictive uncertainty bands are built from the error model.

Therefore, the aim of the research was to emulate previous related papers, mainly research of Schoups and Vrugt (2010) with the modifications proposed by Evin et al. (2013). However, from the beginning of our research, we came across **unexpected problems**. The fact is that, when in a joint inference, the error changing-variance (and/or the changing-bias) is modeled as in Schoups and Vrugt (2010), Evin et al. (2013, 2014), Scharnagl et al. (2015) or ourselves, Total Laws (TLs) enforcement is necessary for their compliance. And compliancy of TLs should not be optional from a strict statistical point of view. From the onset of those unexpected problems, we did not want to find the best error model (neither the best hydrological model) which yields the best performance for the French Broad basin. Hypothesized components of the error model (variance model, bias, model, etc.) were chosen with the main purpose of conducting the true reason behind our manuscript: bringing to the light the statistical requirement of the **Total Laws' fulfillment** and making the recommendation of having particular care about it, when joint inferences with sophisticated error models are performed. Of course, fulfillment of TLs is not enough to achieve reliable inferences, but it is a statistically necessary condition. Besides, suitability of hydrological and error models is also essential in the achievement of this reliability.

**Secondly, in response to the question requested by first reviewer, about the necessity of using a different more "problematic" basin**, our opinion is that we are presenting a mathematical (statistical) problem, and we are dealing with it and proposing one solution, from a mathematical point of view. We have also demonstrated empirically the suitability of our solution with the French Broad basin. There is **one single mathematical result** which evidences that something goes wrong by neglecting TLs in inferences with sophisticated error models, as for example with WLS or GL++:

after its inference, calculation of error (marginal) variance yields a result which does have nothing to do with which would be mathematically expected, that is to say, with the mean of the conditional variances. But, in addition to this mathematical issue, **more examples of another different spurious problems** were shown in the manuscript and/or posted in replies to reviewers, concerning different issues: non-identifiability of error model parameters, meaningless enlargement of the uncertainty band or even the abnormal increment of the correlation between hydrological parameters, shown all of them when TLs are neglected with complex error models (e.g. GL++, GL++bias).

Even so, during this discussion, we have made an important additional effort trying to contribute with another clarifying example of inference, with another different basin. In this case, we have chosen a "problematic" basin which is also within the MOPEX project, with name Guadalupe River. It is the driest basin in the MOPEX experiment, and its challenge was demonstrated in Evin et al. (2014) among many others. Serving as example of how difficult can be its modeling, the GR4J's calibration with SLS error model yields a low Nash-Sutcliffe index (NSE=0.46), which indicates the existence of severe problems in the hydrological model structure and/or in the data. We have performed **the Bayesian joint inference of GR4J's parameters, with the GL++ error model, for Guadalupe basin**, and for the same period considered with French Broad basin. Unsurprisingly, results do not add new important or significant things, that we had not previously shown for the French Broad basin. As an example, in the following figure we show the inferred posteriors with TLs enforcement (GL++ on top) and without it (GL++NTL on bottom).

[Figure]

[Figure]

There are two main things to remark about these two new figures, and both are in the same sense. Firstly, **kappa parameter**, the slope in the linear variance model is identifiable (although it has a high value) when TLs are enforced, but it is not when TLs are neglected (adopts the value of one, which is the established higher bound for this parameter). It is important to note that **this is a new spurious effect**, which did not appear in any previous inference with French Broad basin. Secondly, **autocorrelation parameter $phi_1$** is identifiable (although also with a very high value) when TLs are enforced, but it is not when TLs are neglected, because it adopts the value of one, which does not make sense in an AR(1) model. Effectively, extremely high values for *kappa* and *$phi_1$* are a symptom that Guadalupe is a "difficult" basin (surely as many others) and the generated error structure seems too complex, to be modeled with a simple linear variance model and a first-order autoregressive model, as we and previous researches have made. However, we have been able to identify the full parameter set, but only when we have enforced TLs.

**Thirdly, in response to the question, why do not problems appear in all MOPEX basins?** Schoups and Vrugt (2010) introduced an innovative and generalized formal Bayesian aggregated approach for the error modeling. This sophisticated, but also highly flexible approach is based on the independent modeling of practically all the statistical features of the error conditional distributions: its variance, bias, kurtosis and skewness. Therefore, the Schoups and Vrugt (2010) methodology allows to **characterize the heteroscedasticity, the bias and non-normality** of the hydrological model errors.

So far, the **treatment of the error heteroscedasticity** had been performed through a previous transformation (e.g. **Box-Cox transformations**) of the original variables, with

the aim of stabilizing the non-constant errors variance. Research of Del Giudice et al. (2013) is a good and recent example among many others. Long time ago, Sorooshian and Dracup (1980) were the first who made use of a **direct method** (considering the untransformed original variables) for the error changing-variance modeling. This is a special case of a direct error variance modeling, because their direct method was partially based on a constant error variance, previously obtained from a stabilizing transformation. On balance, their direct methodology established "special" conditions (or restrictions) on the error variance model parameters, for their inference with the untransformed original variables. That is to say, **error variance model parameters were jointly but not freely inferred** with the hydrological ones. After that work and during a prolonged period, nobody (in our knowledge) tried a direct approach for the inference of an error variance model.

Yet recently, the proposed approach by Schoups and Vrugt (2010) also adopted the direct treatment for the error variance modeling. But in their methodology, with a simple linear error variance model (same as ours and as Sorooshian and Dracup (1980)), they did not impose any condition to infer these variance model parameters, differently to Sorooshian and Dracup (1980). Schoups and Vrugt (2010)'s approach, was based on the model specification for the statistical features of the error conditional distributions. In this respect, one of our first questions was, how these error conditional distributions, with practically all its properties varying during the inference, can build their parent bivariate pdf P($e$-$q_s$) distribution? Our belief is that some kind of additional formulation (in essence, a sort of **restriction in the error model parameters**) could be necessary for maintaining the statistical coherence of the whole. When this statistical coherence occurs, Total Laws are fulfilled, or what is the same, **marginal and conditional distributions** are properly related, since both **belong to a unique properly defined bivariate distribution**.

Our opinion is that, the greater or lesser **complexity of the inferred error structure** (mathematically represented by the bivariate pdf P($e$-$q_s$)) seems to predetermine the **occurrence (or not) of the spurious problems** during the joint inference of the variance model, when its parameters are not properly constrained. French Broad basin as an "easy" basin, it has an error structure not too much complex and the problems only appear when 5 unrestricted error model parameters are freely inferred, as shown in our original manuscript: however, as it was shown by Schoups and Vrugt (2010) and in our manuscript (with the restriction of one of the variance model parameters by fulfilling TLs), with 4 freely inferred parameters (and of course also with 3, in Evin et al. (2014)), this basin does not show problems, even if TLs are not considered. On the other hand, basins which are more difficult to model, as Guadalupe basin, yield a more complex error structure whose joint inference does not support, without showing problems, to leave free the variance model parameters. Guadalupe basin showed problems in Evin et al. (2014), with only 3 freely inferred error model parameters (2 for the variance and 1 for the AR (1)). In our inference on Guadalupe, we have not found spurious problems with "five" error model parameters (truly four, if we consider the restriction imposed by TLs for one of the variance model parameters); but as could be expected, since Guadalupe had problems yet with only 3 freely inferred parameters, with five free parameters, we have also found problems. Therefore, the question could be, how many error parameters supports the inference of a basin, without showing

problems, when its error variance model parameters are freely inferred, namely without TLs enforcement? In our opinion, this question does not make sense from a statistical point of view, and it is out of the scope of our paper.

From Schoups and Vrugt (2010)'s, other researches who have performed their same approach for the modeling of the error variance (e.g. Evin et al. (2013, 2014), Scharnagl et al. (2015)), have neglected the statistical necessity of relating marginal and conditional error distributions. This is a statistical pitfall: we realize that this is a hard affirmation, which inevitably does not sound good. But sincerely, we do not find how to word this in a better but also unambiguous manner. Another different thing is that, by neglecting the TLs issue, problems appear or not in the inference process. The underlying statistical incoherence exists by only not fulfilling TLs, although spurious problems do not appear. We recognize the **non-existence of a biunivocal relation** between violation of TLs and the occurrence of problems with the inference, as previous researches have demonstrated, and we have also tested. However, the inexistence of this reciprocity does not mean that those inferences without problems are correct. We do not say that, necessarily they are not correct. We only say that **to be statistically correct, they should comply with TLs**, if not by its own, through some other restrictions or procedure. Our proposed procedure is the TLs enforcement, as explained in our manuscript.

Total Laws could be understood as one method to reduce the degrees of freedom in the inference problem. However, in our opinion, TLs enforcement is more than an ad-hoc method to restrict the error model parameters. Meeting TLs is, from the theoretical point of view, a statistical requirement which eventually produces the convenient error model parameter restriction. Total Laws are fulfilled in classical inferences (as the SLS method) and we understand that, compliance of these laws should be transposed to any inference which involves to the inferred variables and its errors.